# A Randomized Approach to Tight Privacy Accounting

**Jiachen T. Wang**
Princeton University
tianhaowang@princeton.edu

**Saeed Mahloujifar**
Princeton University
sfar@princeton.edu

**Tong Wu**
Princeton University
tongwu@princeton.edu

**Ruoxi Jia**
Virginia Tech
ruoxijia@vt.edu

**Prateek Mittal**
Princeton University
pmittal@princeton.edu

## Abstract

Bounding privacy leakage over compositions, i.e., privacy accounting, is a key challenge in differential privacy (DP). However, the privacy parameter ($\varepsilon$ or $\delta$) is often easy to estimate but hard to bound. In this paper, we propose a new differential privacy paradigm called estimate-verify-release (EVR), which tackles the challenges of providing a strict upper bound for the privacy parameter in DP compositions by converting an *estimate* of privacy parameter into a formal guarantee. The EVR paradigm first verifies whether the mechanism meets the *estimated* privacy guarantee, and then releases the query output based on the verification result. The core component of the EVR is privacy verification. We develop a randomized privacy verifier using Monte Carlo (MC) technique. Furthermore, we propose an MC-based DP accountant that outperforms existing DP accounting techniques in terms of accuracy and efficiency. MC-based DP verifier and accountant is applicable to an important and commonly used class of DP algorithms, including the famous DP-SGD. An empirical evaluation shows the proposed EVR paradigm improves the utility-privacy tradeoff for privacy-preserving machine learning.

## 1   Introduction

The concern of privacy is a major obstacle to deploying machine learning (ML) applications. In response, ML algorithms with differential privacy (DP) guarantees have been proposed and developed. For privacy-preserving ML algorithms, DP mechanisms are often repeatedly applied to private training data. For instance, when training deep learning models using DP-SGD [1], it is often necessary to execute sub-sampled Gaussian mechanisms on the private training data thousands of times.

A major challenge in machine learning with differential privacy is *privacy accounting*, i.e., measuring the privacy loss of the composition of DP mechanisms. A privacy accountant takes a list of mechanisms, and returns the privacy parameter ($\varepsilon$ and $\delta$) for the composition of those mechanisms. Specifically, a privacy accountant is given a target $\varepsilon$ and finds the *smallest achievable* $\delta$ such that the composed mechanism $\mathcal{M}$ is ($\varepsilon, \delta$)-DP (we can also fix $\delta$ and find $\varepsilon$). We use $\delta_{\mathcal{M}}(\varepsilon)$ to denote the smallest achievable $\delta$ given $\varepsilon$, which is often referred to as *optimal privacy curve* in the literature.

Training deep learning models with DP-SGD is essentially the *adaptive composition* for thousands of sub-sampled Gaussian Mechanisms. Moment Accountant (MA) is a pioneer solution for privacy loss calculation in differentially private deep learning [1]. However, MA does not provide the optimal $\delta_{\mathcal{M}}(\varepsilon)$ in general [44]. This motivates the development of more advanced privacy accounting techniques that outperforms MA. Two major lines of such works are based on Fast Fourier Transform (FFT) (e.g., [19]) and Central Limit Theorem (CLT) [7, 41]. Both techniques can provide an *estimate*

37th Conference on Neural Information Processing Systems (NeurIPS 2023).

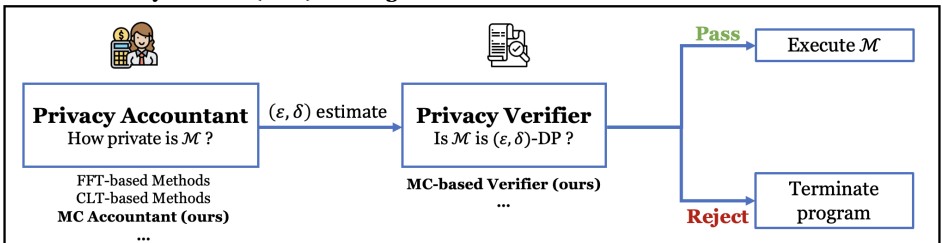

**Estimate-Verify-Release (EVR) Paradigm**

Figure 2: An overview of our EVR paradigm. EVR converts an estimated $(\varepsilon, \delta)$ provided by a privacy accountant into a formal guarantee. Compared with the original mechanism, the EVR has an extra failure mode that does not output anything when the estimated $(\varepsilon, \delta)$ is rejected. We show that the MC-based verifier we proposed can achieve negligible failure probability $(O(\delta))$ in Section 4.4.

as well as an upper bound for $\delta_{\mathcal{M}}(\varepsilon)$ though bounding the worst-case estimation error. In practice, **only the upper bounds for $\delta_{\mathcal{M}}(\varepsilon)$ can be used**, as differential privacy is a strict guarantee.

**Motivation: estimates can be more accurate than upper bounds.** The motivation for this paper stems from the limitations of current privacy accounting techniques in providing tight upper bounds for $\delta_{\mathcal{M}}(\varepsilon)$. Despite outperforming MA, both FFT- and CLT-based methods can provide ineffective bounds in certain regimes [19, 41]. We demonstrate such limitations in Figure 1 using the composition of Gaussian mechanisms. For FFT-based technique [19], we can see that although it outperforms MA for most of the regimes, the upper bounds (blue dashed curve) are worse than that of MA when $\delta < 10^{-10}$ due to computational limitations (as discussed in [19]'s Appendix A; also see Remark 6 for a discussion of why the regime of $\delta < 10^{-10}$ is important). The CLT-based techniques (e.g., [41]) also produce sub-optimal upper bounds (red dashed curve) for the entire range of $\delta$. This is primarily due to the small number of mechanisms used ($k = 1200$), which does not meet the requirements for CLT bounds to converge (similar phenomenon observed in [41]). On the other hand, we can see that the *estimate*s of $\delta_{\mathcal{M}}(\varepsilon)$ from both FFT and CLT-based techniques, which estimate the parameters rather than providing an upper bound, are in fact very close to the ground truth (the three curves overlapped in Figure 1). However, as we mentioned earlier, these accurate estimations cannot be used in practice, as we cannot prove that they do not underestimate $\delta_{\mathcal{M}}(\varepsilon)$. The

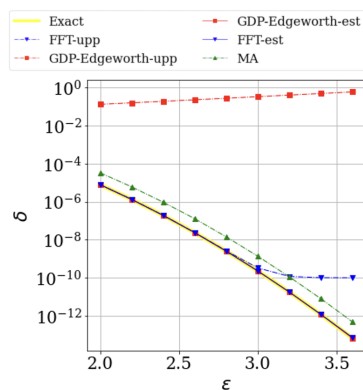

Figure 1: Results of estimating/bounding $\delta_{\mathcal{M}}(\varepsilon)$ for the composition of 1200 Gaussian mechanisms with $\sigma = 70$. '-upp' means upper bound and '-est' means estimate. Curves of 'Exact', 'FFT-est', and 'CLT-est' are overlapped. The groundtruth curve ('Exact') for pure Gaussian mechanism can be computed analytically [19].

dilemma raises an important question: *can we develop new techniques that allow us to use privacy parameter estimates instead of strict upper bounds in privacy accounting?*[1]

This paper gives a positive answer to it. Our contributions are summarized as follows:

**Estimate-Verify-Release (EVR): a DP paradigm that converts privacy parameter estimate into a formal guarantee.** We develop a new DP paradigm called *Estimate-Verify-Release*, which augments a mechanism with a formal privacy guarantee based on its privacy parameter estimates. The basic idea of EVR is to first verify whether the mechanism satisfies the estimated DP guarantee, and release the mechanism's output if the verification is passed. The core component of the EVR paradigm is **privacy verification**. A DP verifier can be randomized and imperfect, suffering from both false positives (accept an underestimation) and false negatives (reject an overestimation). We show that EVR's privacy guarantee can be achieved when privacy verification has a low false negative rate.

**A Monte Carlo-based DP Verifier.** For an important and widely used class of DP algorithms including Subsampled Gaussian mechanism (the building block for DP-SGD), we develop a Monte

---

[1]We note that this not only brings benefits for the regime where $\delta < 10^{-10}$, but also for the more common regime where $\delta \approx 10^{-5}$. See Figure 3 for an example.

Carlo (MC) based DP verifier for the EVR paradigm. We present various techniques that ensure the DP verifier has both a low false positive rate (for privacy guarantee) and a low false negative rate (for utility guarantee, i.e., making the EVR and the original mechanism as similar as possible).

**A Monte Carlo-based DP Accountant.** We further propose a new MC-based approach for DP accounting, which we call the *MC accountant*. It utilizes similar MC techniques as in privacy verification. We show that the MC accountant achieves several advantages over existing privacy accounting methods. In particular, we demonstrate that MC accountant is efficient for *online privacy accounting*, a realistic scenario for privacy practitioners where one wants to update the estimate on privacy guarantee whenever executing a new mechanism.

Figure 2 gives an overview of the proposed EVR paradigm as well as this paper's contributions.

## 2 Privacy Accounting: a Mean Estimation/Bounding Problem

In this section, we review relevant concepts and introduce privacy accounting as a mean estimation/bounding problem.

**Symbols and notations.** We use $D, D' \in \mathbb{N}^{\mathcal{X}}$ to denote two datasets with an unspecified size over space $\mathcal{X}$. We call two datasets $D$ and $D'$ *adjacent* (denoted as $D \sim D'$) if we can construct one by adding/removing one data point from the other. We use $P, Q$ to denote random variables. We also overload the notation and denote $P(\cdot), Q(\cdot)$ the density function of $P, Q$.

**Differential privacy and its equivalent characterizations.** Having established the notations, we can now proceed to formally define differential privacy.

**Definition 1** (Differential Privacy [13]). *For $\varepsilon, \delta \geq 0$, a randomized algorithm $\mathcal{M} : \mathbb{N}^{\mathcal{X}} \to \mathcal{Y}$ is $(\varepsilon, \delta)$-differentially private if for every pair of adjacent datasets $D \sim D'$ and for every subset of possible outputs $E \subseteq \mathcal{Y}$, we have $\Pr_{\mathcal{M}}[\mathcal{M}(D) \in E] \leq e^{\varepsilon} \Pr_{\mathcal{M}}[\mathcal{M}(D') \in E] + \delta$.*

One can alternatively define differential privacy in terms of the maximum possible divergence between the output distribution of any pair of $\mathcal{M}(D)$ and $\mathcal{M}(D')$.

**Lemma 2** ([5]). *A mechanism $\mathcal{M}$ is $(\varepsilon, \delta)$-DP iff $\sup_{D \sim D'} \mathsf{E}_{e^{\varepsilon}}(\mathcal{M}(D) \| \mathcal{M}(D')) \leq \delta$, where $\mathsf{E}_{\gamma}(P \| Q) := \mathbb{E}_{o \sim Q}[(\frac{P(o)}{Q(o)} - \gamma)_+]$ & $(a)_+ := \max(a, 0)$.*

$\mathsf{E}_{\gamma}$ is usually referred as *Hockey-Stick* (HS) Divergence in the literature. For every mechanism $\mathcal{M}$ and every $\varepsilon \geq 0$, there exists a smallest $\delta$ such that $\mathcal{M}$ is $(\varepsilon, \delta)$-DP. Following the literature [44, 2], we formalize such a $\delta$ as a function of $\varepsilon$.

**Definition 3** (Optimal Privacy Curve). *The* optimal privacy curve *of a mechanism $\mathcal{M}$ is the function $\delta_{\mathcal{M}} : \mathbb{R}^+ \to [0, 1]$ s.t. $\delta_{\mathcal{M}}(\varepsilon) := \sup_{D \sim D'} \mathsf{E}_{e^{\varepsilon}}(\mathcal{M}(D) \| \mathcal{M}(D'))$.*

**Dominating Distribution Pair and Privacy Loss Random Variable (PRV).** It is computationally infeasible to find $\delta_{\mathcal{M}}(\varepsilon)$ by computing $\mathsf{E}_{e^{\varepsilon}}(\mathcal{M}(D) \| \mathcal{M}(D'))$ for all pairs of adjacent dataset $D$ and $D'$. A mainstream strategy in the literature is to find a pair of distributions $(P, Q)$ that dominates all $(\mathcal{M}(D), \mathcal{M}(D'))$ in terms of the Hockey-Stick divergence. This results in the introduction of *dominating distribution pair* and *privacy loss random variable* (PRV).

**Definition 4** ([44]). *A pair of distributions $(P, Q)$ is a pair of* dominating distributions *for $\mathcal{M}$ under adjacent relation $\sim$ if* **for all** *$\gamma \geq 0$, $\sup_{D \sim D'} \mathsf{E}_{\gamma}(\mathcal{M}(D) \| \mathcal{M}(D')) \leq \mathsf{E}_{\gamma}(P \| Q)$. If equality is achieved* **for all** *$\gamma \geq 0$, then we say $(P, Q)$ is a pair of* tightly *dominating distributions for $\mathcal{M}$. Furthermore, we call $Y := \log\left(\frac{P(o)}{Q(o)}\right), o \sim P$ the* privacy loss random variable (PRV) *of $\mathcal{M}$ associated with dominating distribution pair $(P, Q)$.*

Zhu et al. [44] shows that all mechanisms have a pair of tightly dominating distributions. Hence, we can alternatively characterize the optimal privacy curve as $\delta_{\mathcal{M}}(\varepsilon) = \mathsf{E}_{e^{\varepsilon}}(P \| Q)$ for the tightly dominating pair $(P, Q)$, and we have $\delta_{\mathcal{M}}(\varepsilon) \leq \mathsf{E}_{e^{\varepsilon}}(P \| Q)$ if $(P, Q)$ is a dominating pair that is not necessarily tight. The importance of the concept of PRV comes from the fact that we can write $\mathsf{E}_{e^{\varepsilon}}(P \| Q)$ as an expectation over it: $\mathsf{E}_{e^{\varepsilon}}(P \| Q) = \mathbb{E}_Y\left[\left(1 - e^{\varepsilon - Y}\right)_+\right]$. Thus, one can bound $\delta_{\mathcal{M}}(\varepsilon)$ by first identifying $\mathcal{M}$'s dominating pair distributions as well as the associated PRV $Y$, and then computing this expectation. Such a formulation allows us to bound $\delta_{\mathcal{M}}(\varepsilon)$ without enumerating over

all adjacent $D$ and $D'$. For notation convenience, we denote $\delta_Y(\varepsilon) := \mathbb{E}_Y \left[ \left( 1 - e^{\varepsilon - Y} \right)_+ \right]$. Clearly, $\delta_\mathcal{M} \leq \delta_Y$. If $(P, Q)$ is a tightly dominating pair for $\mathcal{M}$, then $\delta_\mathcal{M} = \delta_Y$.

**Privacy Accounting as a Mean Estimation/Bounding Problem.** Privacy accounting aims to estimate and bound the optimal privacy curve $\delta_\mathcal{M}(\varepsilon)$ for adaptively composed mechanism $\mathcal{M} = \mathcal{M}_1 \circ \cdots \circ \mathcal{M}_k(D)$. The *adaptive composition* of two mechanisms $\mathcal{M}_1$ and $\mathcal{M}_2$ is defined as $\mathcal{M}_1 \circ \mathcal{M}_2(D) := (\mathcal{M}_1(D), \mathcal{M}_2(D, \mathcal{M}_1(D)))$, in which $\mathcal{M}_2$ can access both the dataset and the output of $\mathcal{M}_1$. Most of the practical privacy accounting techniques are based on the concept of PRV, centered on the following result.

**Lemma 5** ([44]). *Let $(P_j, Q_j)$ be a pair of tightly dominating distributions for mechanism $\mathcal{M}_j$ for $j \in \{1, \ldots, k\}$. Then $(P_1 \times \cdots \times P_k, Q_1 \times \cdots \times Q_k)$ is a pair of dominating distributions for $\mathcal{M} = \mathcal{M}_1 \circ \cdots \circ \mathcal{M}_k$, where $\times$ denotes the product distribution. Furthermore, the associated privacy loss random variable is $Y = \sum_{i=1}^{k} Y_i$ where $Y_i$ is the PRV associated with $(P_i, Q_i)$.*

Lemma 5 suggests that privacy accounting for DP composition can be cast into a *mean estimation/bounding problem* where one aims to approximate or bound the expectation in (2) when $Y = \sum_{i=1}^{k} Y_i$. Note that while Lemma 5 does not guarantee a pair of tightly dominating distributions for the adaptive composition, it cannot be improved in general, as noted in [10]. Hence, all the current privacy accounting techniques work on $\delta_Y$ instead of $\delta_\mathcal{M}$, as Lemma 5 is tight even for non-adaptive composition. Following the prior works, in this paper, we only consider the practical scenarios where Lemma 5 is tight for the simplicity of presentation. That is, we assume $\delta_Y = \delta_\mathcal{M}$ unless otherwise specified.

Most of the existing privacy accounting techniques can be described as different techniques for such a mean estimation problem. **Example-1: FFT-based methods.** This line of works (e.g., [19]) discretizes the domain of each $Y_i$ and use Fast Fourier Transform (FFT) to speed up the approximation of $\delta_Y(\varepsilon)$. The upper bound is derived through the worst-case error bound for the approximation.
**Example-2: CLT-based methods.** [7, 41] use CLT to approximate the distribution of $Y = \sum_{i=1}^{k} Y_i$ as Gaussian distribution. They then use CLT's finite-sample approximation guarantee to derive the upper bound for $\delta_Y(\varepsilon)$.

**Remark 6** (**The Importance of Privacy Accounting in Regime** $\delta < 10^{-10}$). *The regime where $\delta < 10^{-10}$ is of significant importance for two reasons. (1) $\delta$ serves as an upper bound on the chance of severe privacy breaches, such as complete dataset exposure, necessitating a "cryptographically small" value, namely, $\delta < n^{-\omega(1)}$ [14, 40]. (2) Even with the oft-used yet questionable guideline of $\delta \approx n^{-1}$ or $n^{-1.1}$, datasets of modern scale, such as JFT-3B [43] or LAION-5B [37], already comprise billions of records, thus rendering small $\delta$ values crucial. While we acknowledge that it requires a lot of effort to achieve a good privacy-utility tradeoff even for the current choice of $\delta \approx n^{-1}$, it is important to keep such a goal in mind.*

# 3 Estimate-Verify-Release

As mentioned earlier, upper bounds for $\delta_Y(\varepsilon)$ are the only valid options for privacy accounting techniques. However, as we have demonstrated in Figure 1, both FFT- and CLT-based methods can provide overly conservative upper bounds in certain regimes. On the other hand, their *estimates* for $\delta_Y(\varepsilon)$ can be very close to the ground truth even though there is no provable guarantee. Therefore, it is highly desirable to develop new techniques that enable the use of privacy parameter estimates instead of overly conservative upper bounds in privacy accounting.

We tackle the problem by introducing a new paradigm for constructing DP mechanisms, which we call Estimate-Verify-Release (EVR). The key component of the EVR is an object called DP verifier (Section 3.1). The full EVR paradigm is then presented in Section 3.2, where the DP verifier is utilized as a building block to guarantee privacy.

## 3.1 Differential Privacy Verifier

We first formalize the concept of *differential privacy verifier*, the central element of the EVR paradigm. In informal terms, a DP verifier is an algorithm that attempts to verify whether a mechanism satisfies a specific level of differential privacy.

**Definition 7** (Differential Privacy Verifier). *We say a differentially private verifier $DPV(\cdot)$ is an algorithm that takes the description of a mechanism $\mathcal{M}$ and proposed privacy parameter $(\varepsilon, \delta^{\mathrm{est}})$ as input, and returns $\mathtt{True} \leftarrow DPV(\mathcal{M}, \varepsilon, \delta^{\mathrm{est}})$ if the algorithm believes $\mathcal{M}$ is $(\varepsilon, \delta^{\mathrm{est}})$-DP (i.e., $\delta^{\mathrm{est}} \geq \delta_Y(\varepsilon)$ where $Y$ is the PRV of $\mathcal{M}$), and returns $\mathtt{False}$ otherwise.*

A differential privacy verifier can be imperfect, suffering from both false positives (FP) and false negatives (FN). Typically, FP rate is the likelihood for DPV to accept $(\varepsilon, \delta^{\mathrm{est}})$ when $\delta^{\mathrm{est}} < \delta_Y(\varepsilon)$. However, $\delta^{\mathrm{est}}$ is still a good estimate for $\delta_Y(\varepsilon)$ by being a small (e.g., <10%) underestimate. To account for this, we introduce a smoothing factor, $\tau \in (0, 1]$, such that $\delta^{\mathrm{est}}$ is deemed "should be rejected" only when $\delta^{\mathrm{est}} \leq \tau \delta_Y(\varepsilon)$. A similar argument can be put forth for FN cases where we also introduce a smoothing factor $\rho \in (0, 1]$. This leads to relaxed notions for FP/FN rate:

**Definition 8.** *We say a DPV's $\tau$-relaxed false positive rate at $(\varepsilon, \delta^{\mathrm{est}})$ is*

$$\mathsf{FP}_{DPV}(\varepsilon, \delta^{\mathrm{est}}; \tau) := \sup_{\mathcal{M}: \delta^{\mathrm{est}} < \tau \delta_Y(\varepsilon)} \Pr_{DPV}\left[DPV(\mathcal{M}, \varepsilon, \delta^{\mathrm{est}}) = \mathtt{True}\right]$$

*We say a DPV's $\rho$-relaxed false negative rate at $(\varepsilon, \delta^{\mathrm{est}})$ is*

$$\mathsf{FN}_{DPV}(\varepsilon, \delta^{\mathrm{est}}; \rho) := \sup_{\mathcal{M}: \delta^{\mathrm{est}} > \rho \delta_Y(\varepsilon)} \Pr_{DPV}\left[DPV(\mathcal{M}, \varepsilon, \delta^{\mathrm{est}}) = \mathtt{False}\right]$$

**Privacy Verification with DP Accountant.** For a composed mechanism $\mathcal{M} = \mathcal{M}_1 \circ \ldots \circ \mathcal{M}_k$, a DP verifier can be easily implemented using any existing privacy accounting techniques. That is, one can execute DP accountant to obtain an estimate or upper bound $(\varepsilon, \hat{\delta})$ of the actual privacy parameter. If $\delta^{\mathrm{est}} < \hat{\delta}$, then the proposed privacy level is rejected as it is more private than what the DP accountant tells; otherwise, the test is passed. The input description of a mechanism $\mathcal{M}$, in this case, can differ depending on the DP accounting method. For Moment Accountant [1], the input description is the upper bound of the moment-generating function (MGF) of the privacy loss random variable for each individual mechanism. For FFT and CLT-based methods, the input description is the cumulative distribution functions (CDF) of the dominating distribution pair of each individual $\mathcal{M}_i$.

## 3.2 EVR: Ensuring Estimated Privacy with DP Verifier

We now present the full paradigm of EVR. As suggested by the name, it contains three steps: **(1) Estimate:** A privacy parameter $(\varepsilon, \delta^{\mathrm{est}})$ for $\mathcal{M}$ is estimated, e.g., based on a privacy auditing or accounting technique. **(2) Verify:** A DP verifier DPV is used for validating whether mechanism $\mathcal{M}$ sat-

---

**Algorithm 1** Estimate-Verify-Release (EVR) Framework

1: **Input:** $\mathcal{M}$: mechanism. $D$: dataset. $(\varepsilon, \delta^{\mathrm{est}})$: an estimated privacy parameter for $\mathcal{M}$.
2: **if** $\mathrm{DPV}(\mathcal{M}, \varepsilon, \delta^{\mathrm{est}})$ outputs $\mathtt{True}$ **then** Execute $\mathcal{M}(D)$.
3: **else** Print $\perp$.

---

isfies $(\varepsilon, \delta^{\mathrm{est}})$-DP guarantee. **(3) Release:** If DP verification test is passed, we can execute $\mathcal{M}$ as usual; otherwise, the program is terminated immediately. For practical utility, this rejection probability needs to be small when $(\varepsilon, \delta^{\mathrm{est}})$ is an accurate estimation. The procedure is summarized in Algorithm 1.

Given estimated privacy parameter $(\varepsilon, \delta^{\mathrm{est}})$, we have the privacy guarantee for the EVR paradigm:

**Theorem 9.** *Algorithm 1 is $(\varepsilon, \delta^{\mathrm{est}}/\tau)$-DP for any $\tau > 0$ if $\mathsf{FP}_{DPV}(\varepsilon, \delta^{\mathrm{est}}; \tau) \leq \delta^{\mathrm{est}}/\tau$.*

We defer the proof to Appendix B. The implication of this result is that, for any *estimate* of the privacy parameter, one can safely use it as a DPV with a bounded false positive rate would enforce differential privacy. However, this is not enough: an overly conservative DPV that satisfies 0 FP rate but rejects everything would not be useful. When $\delta^{\mathrm{est}}$ is accurate, we hope the DPV can also achieve a small *false negative rate* so that the output distributions of EVR and $\mathcal{M}$ are indistinguishable. We discuss the instantiation of DPV in Section 4.

## 4 Monte Carlo Verifier of Differential Privacy

As we can see from Section 3.2, a DP verifier (DPV) that achieves a small FP rate is the central element for the EVR framework. In the meanwhile, it is also important that DPV has a low FN rate in

order to maintain the good utility of the EVR when the privacy parameter estimate is accurate. In this section, we introduce an instantiation of DPV based on the Monte Carlo technique that achieves both a low FP and FN rate, assuming the PRV is known for each individual mechanism.

**Remark 10** (**Mechanisms where PRV can be derived**). *PRV can be derived for many commonly used DP mechanisms such as the Laplace, Gaussian, and Subsampled Gaussian Mechanism [24, 19]. In particular, our DP verifier applies for DP-SGD, one of the most important application scenarios of privacy accounting. Moreover, the availability of PRV is also the assumption for most of the recently developed privacy accounting techniques (including FFT- and CLT-based methods). The extension beyond these commonly used mechanisms is an important future work in the field.*

**Remark 11** (**Previous studies on the hardness of privacy verification**). *Several studies [16, 8] have shown that DP verification is an NP-hard problem. However, these works consider the setting where the input description of the DP mechanism is its corresponding randomized Boolean circuits. Some other works [18] show that DP verification is impossible, but this assertion is proved for the black-box setting where the verifier can only query the mechanism. Our work gets around this barrier by providing the description of the PRV of the mechanism as input to the verifier.*

## 4.1  DPV **through an MC Estimator for** $\delta_Y(\varepsilon)$

Recall that most of the recently proposed DP accountants are essentially different techniques for estimating the expectation

$$\delta_{Y=\sum_{i=1}^k Y_i}(\varepsilon) = \mathbb{E}_Y\left[\left(1 - e^{\varepsilon - Y}\right)_+\right]$$

where each $Y_i$ is the privacy loss random variable $Y_i = \log\left(\frac{P_i(t)}{Q_i(t)}\right)$ for $t \sim P_i$, and $(P_i, Q_i)$ is a pair of dominating distribution for individual mechanism $\mathcal{M}_i$. In the following text, we denote the product distribution $\boldsymbol{P} := P_1 \times \ldots \times P_k$ and $\boldsymbol{Q} := Q_1 \times \ldots \times Q_k$. Recall from Lemma 5 that $(\boldsymbol{P}, \boldsymbol{Q})$ is a pair of dominating distributions for the composed mechanism $\mathcal{M}$. For notation simplicity, we denote a vector $\boldsymbol{t} := (t^{(1)}, \ldots, t^{(k)})$.

*Monte Carlo* (MC) technique is arguably one of the most natural and widely used techniques for approximating expectations. Since $\delta_Y(\varepsilon)$ is an expectation in terms of the PRV $Y$, one can apply MC-based technique to estimate it. Given an MC estimator for $\delta_Y(\varepsilon)$, we construct a DPV$(\mathcal{M}, \varepsilon, \delta^{\text{est}})$ as shown in Algorithm 2 (instantiated by the Simple MC estimator introduced in Section 4.2). Specifically, we first obtain an estimate $\widehat{\delta}$ from an MC estimator for $\delta_Y(\varepsilon)$. The estimate $\delta^{\text{est}}$ passes the test if

---

**Algorithm 2**  DPV$(\mathcal{M}, \varepsilon, \delta^{\text{est}})$ with Simple MC Estimator and Offset Parameter $\Delta$.

1:  Obtain i.i.d. samples $\{\boldsymbol{t}_i\}_{i=1}^m$ from $\boldsymbol{P}$.
2:  Compute $\widehat{\delta} = \frac{1}{m}\sum_{i=1}^m (1 - e^{\varepsilon - y_i})_+$ with PRV samples $y_i = \log\left(\frac{\boldsymbol{P}(\boldsymbol{t}_i)}{\boldsymbol{Q}(\boldsymbol{t}_i)}\right), i = 1 \ldots m$.
3:  **if** $\widehat{\delta} < \frac{\delta^{\text{est}}}{\tau} - \Delta$ **then** return True.
4:  **else** return False.

---

$\widehat{\delta} < \frac{\delta^{\text{est}}}{\tau} - \Delta$, and fails otherwise. The parameter $\Delta \geq 0$ here is an offset that allows us to conveniently controls the $\tau$-relaxed false positive rate. We will discuss how to set $\Delta$ in Section 4.4.

In the following contents, we first present two constructions of MC estimators for $\delta_Y(\varepsilon)$ in Section 4.2. We then discuss the condition for which our MC-based DPV achieves a certain target FP rate in Section 4.3. Finally, we discuss the utility guarantee for the MC-based DPV in Section 4.4.

## 4.2  **Constructing MC Estimator for** $\delta_Y(\varepsilon)$

In this section, we first present a simple MC estimator that applies to any mechanisms where we can derive and sample from the dominating distribution pairs. Given the importance of Poisson Subsampled Gaussian mechanism for privacy-preserving machine learning, we further design a more advanced and specialized MC estimator for it based on the importance sampling technique.

**Simple Monte Carlo Estimator.** One can easily sample from $Y$ by sampling $\boldsymbol{t} \sim \boldsymbol{P}$ and output $\log\left(\frac{\boldsymbol{P}(\boldsymbol{t})}{\boldsymbol{Q}(\boldsymbol{t})}\right)$. Hence, a straightforward algorithm for estimating (2) is the Simple Monte Carlo (SMC) algorithm, which directly samples from the privacy random variable $Y$. We formally define it here.

**Definition 12** (Simple Monte Carlo (SMC) Estimator)**.** *We denote $\widehat{\delta}_{MC}^m(\varepsilon)$ as the random variable of SMC estimator for $\delta_Y(\varepsilon)$ with $m$ samples, i.e., $\widehat{\delta}_{MC}^m(\varepsilon) := \frac{1}{m}\sum_{i=1}^m (1 - e^{\varepsilon - y_i})_+$ for $y_1, \ldots, y_m$ i.i.d. sampled from $Y$.*

**Importance Sampling Estimator for Poisson Subsampled Gaussian (Overview).** As $\delta_Y(\varepsilon)$ is usually a tiny value ($10^{-5}$ or even cryptographically small), it is likely that by naive sampling from $Y$, almost all of the samples in $\{(1 - e^{\varepsilon - y_i})_+\}_{i=1}^m$ are just 0s! That is, the i.i.d. samples $\{y_i\}_{i=1}^m$ from $Y$ can rarely exceed $\varepsilon$. To further improve the sample efficiency, one can potentially use more advanced MC techniques such as Importance Sampling or MCMC. However, these advanced tools usually require additional distributional information about $Y$ and thus need to be developed case-by-case.

Poisson Subsampled Gaussian mechanism is the main workhorse behind the DP-SGD algorithm [1]. Given its important role in privacy-preserving ML, we derive an advanced MC estimator for it based on the Importance Sampling technique. Importance Sampling (IS) is a classic method for rare event simulation [39]. It samples from an alternative distribution instead of the distribution of the quantity of interest, and a weighting factor is then used for correcting the difference between the two distributions. The specific design of alternative distribution is complicated and notation-heavy, and we defer the technical details to Appendix C. At a high level, we construct the alternative sampling distribution based on the *exponential tilting* technique and derive the optimal tilting parameter such that the corresponding IS estimator approximately achieves the smallest variance. Similar to Definition 12, we use $\widehat{\delta}_{IS}^m$ to denote the random variable of importance sampling estimator with $m$ samples.

### 4.3 Bounding FP Rate

We now discuss the FP guarantee for the DPV instantiated by $\widehat{\delta}_{MC}^m$ and $\widehat{\delta}_{IS}^m$ we developed in the last section. Since both estimators are unbiased, by Law of Large Number, both $\widehat{\delta}_{MC}^m$ and $\widehat{\delta}_{IS}^m$ converge to $\delta_Y(\varepsilon)$ almost surely as $m \to \infty$, which leads a DPV with perfect accuracy. Of course, $m$ cannot go to $\infty$ in practice. In the following, we derive the required amount of samples $m$ for ensuring that $\tau$-relaxed false positive rate is smaller than $\delta^{\text{est}}/\tau$ for $\widehat{\delta}_{MC}^m$ and $\widehat{\delta}_{IS}^m$. We use $\widehat{\delta}_{MC}$ (or $\widehat{\delta}_{IS}$) as an abbreviation for $\widehat{\delta}_{MC}^1$ (or $\widehat{\delta}_{IS}^1$), the random variable for a single draw of sampling. We state the theorem for $\widehat{\delta}_{MC}^m$, and the same result for $\widehat{\delta}_{IS}^m$ can be obtained by simply replacing $\widehat{\delta}_{MC}$ with $\widehat{\delta}_{IS}$. We use $\mathsf{FP}_{MC}$ to denote the FP rate for DPV implemented by SMC estimator.

**Theorem 13.** *Suppose $\mathbb{E}\left[\left(\widehat{\delta}_{MC}\right)^2\right] \leq \nu$. DPV instantiated by $\widehat{\delta}_{MC}^m$ has bounded $\tau$-relaxed false positive rate $\mathsf{FP}_{MC}(\varepsilon, \delta^{\text{est}}; \tau) \leq \delta^{\text{est}}/\tau$ with $m \geq \frac{2\nu}{\Delta^2}\log(\tau/\delta^{\text{est}})$.*

The proof is based on Bennett's inequality and is deferred to Appendix D. This result suggests that, to improve the computational efficiency of MC-based DPV (i.e., tighten the number of required samples), it is important to tightly bound $\mathbb{E}[(\widehat{\delta}_{MC})^2]$ (or $\mathbb{E}[(\widehat{\delta}_{IS,\theta})^2]$), the second moment of $\widehat{\delta}_{MC}$ (or $\widehat{\delta}_{IS}$).

**Bounding the Second-Moment of MC Estimators (Overview).** For clarity, we defer the notation-heavy results and derivation of the upper bounds for $\mathbb{E}[(\widehat{\delta}_{MC})^2]$ and $\mathbb{E}[(\widehat{\delta}_{IS})^2]$ to Appendix E. Our high-level idea for bounding $\mathbb{E}[(\widehat{\delta}_{MC})^2]$ is through the RDP guarantee for the composed mechanism $\mathcal{M}$. This is a natural idea since converting RDP to upper bounds for $\delta_Y(\varepsilon)$ – the first moment of $\widehat{\delta}_{MC}$ – is a well-studied problem [30, 9, 3]. Bounding $\mathbb{E}[(\widehat{\delta}_{IS})^2]$ is highly technically involved.

### 4.4 Guaranteeing Utility

**Overall picture so far.** Given the proposed privacy parameter $(\varepsilon, \delta^{\text{est}})$, a tolerable degree of underestimation $\tau$, and an offset parameter $\Delta$, one can now compute the number of samples $m$ required for the MC-based DPV such that $\tau$-relaxed FP rate to be $\leq \delta^{\text{est}}/\tau$ based on the results from Section 4.3 and Appendix E. We have not yet discussed the selection of the hyperparameter $\Delta$. An appropriate $\Delta$ is important for the utility of MC-based DPV. That is, when $\delta^{\text{est}}$ is not too smaller than $\delta_Y(\varepsilon)$, the probability of being rejected by DPV should stay negligible. If we set $\Delta \to \infty$, the DPV simply rejects everything, which achieves 0 FP rate (and with $m = 0$) but is not useful at all!

Formally, the utility of a DPV is quantified by the $\rho$-relaxed false negative (FN) rate (Definition 8). While one may be able to bound the FN rate through concentration inequalities, a more convenient

way is to pick an appropriate $\Delta$ such that $\mathsf{FN}_{\mathrm{DPV}}$ is *approximately* smaller than $\mathsf{FP}_{\mathrm{DPV}}$. After all, $\mathsf{FP}_{\mathrm{DPV}}$ already has to be a small value $\leq \delta^{\mathrm{est}}/\tau$ for privacy guarantee. The result is stated informally in the following (holds for both $\widehat{\delta}_{\mathrm{MC}}$ and $\widehat{\delta}_{\mathrm{IS}}$), and the involved derivation is deferred to Appendix F.

**Theorem 14** (Informal). *When $\Delta = 0.4\,(1/\tau - 1/\rho)\,\delta^{\mathrm{est}}$, then $\mathsf{FN}_{\mathit{MC}}(\varepsilon, \delta^{\mathrm{est}}; \rho) \lesssim \mathsf{FP}_{\mathit{MC}}(\varepsilon, \delta^{\mathrm{est}}; \tau)$.*

Therefore, by setting $\Delta = 0.4\,(1/\tau - 1/\rho)\,\delta^{\mathrm{est}}$, one can ensure that $\mathsf{FN}_{\mathit{MC}}(\varepsilon, \delta^{\mathrm{est}}; \rho)$ is also (approximately) upper bounded by $\Theta(\delta^{\mathrm{est}}/\tau)$. Moreover, in Appendix, we empirically show that the FP rate is actually a very conservative bound for the FN rate. Both $\tau$ and $\rho$ are selected based on the tradeoff between privacy, utility, and efficiency.

The pseudocode of privacy verification for DP-SGD is summarized in Appendix G.

## 5 Monte Carlo Accountant of Differential Privacy

The Monte Carlo estimators $\widehat{\delta}_{\mathrm{MC}}$ and $\widehat{\delta}_{\mathrm{IS}}$ described in Section 4.2 are used for implementing DP verifiers. One may already realize that the same estimators can also be utilized to directly implement a DP accountant which *estimates* $\delta_Y(\varepsilon)$. It is important to note that with the EVR paradigm, DP accountants are no longer required to derive a strict upper bound for $\delta_Y(\varepsilon)$. We refer to the technique of estimating $\delta_Y(\varepsilon)$ using the MC estimators as *Monte Carlo accountant*.

**Finding $\varepsilon$ for a given $\delta$.** It is straightforward to implement MC accountant when we fix $\varepsilon$ and compute for $\delta_Y(\varepsilon)$. In practice, privacy practitioners often want to do the inverse: finding $\varepsilon$ for a given $\delta$, which we denote as $\varepsilon_Y(\delta)$. Similar to the existing privacy accounting methods, we use binary search to find $\varepsilon_Y(\delta)$ (see Algorithm 3). Specifically, after generating PRV samples $\{y_i\}_{i=1}^m$, we simply need to find the $\varepsilon$ such that $\frac{1}{m}\sum_{i=1}^m (1 - e^{\varepsilon - y_i})_+ = \delta$. We do **not**

---

**Algorithm 3** MC Accountant for $\varepsilon_Y(\delta)$.

1: Obtain PRV samples $\{y_i\}_{i=1}^m$ with either Simple MC or Importance Sampling.
2: Binary search $\varepsilon$ such that $\frac{1}{m}\sum_{i=1}^m (1 - e^{\varepsilon - y_i})_+ = \delta$.
3: Return $\varepsilon$.

---

need to generate new PRV samples for different $\varepsilon$ we evaluate during the binary search; hence the additional binary search is computationally efficient.

**Number of Samples for MC Accountant.** Compared with the number of samples required for achieving the FP guarantee in Section 4.3, one may be able to use much fewer samples to obtain a decent estimate for $\delta_Y(\varepsilon)$, as the sample complexity bound derived based on concentration inequality may be conservative. Many heuristics for guiding the number of samples in MC simulation have been developed (e.g., Wald confidence interval) and can be applied to the setting of MC accountants.

Compared with FFT-based and CLT-based methods, MC accountant exhibits the following strength:

**(1) Accurate $\delta_Y(\varepsilon)$ estimation in all regimes.** As we mentioned earlier, the state-of-the-art FFT-based method [19] fails to provide meaningful bounds due to computational limitations when the true value of $\delta_Y(\varepsilon)$ is small. In contrast, the simplicity of the MC accountant allows us to accurately estimate $\delta_Y(\varepsilon)$ in all regimes.

**(2) Short clock runtime & Easy GPU acceleration.** MC-based techniques are well-suited for parallel computing and GPU acceleration due to their nature of repeated sampling. One can easily utilize PyTorch's CUDA functionality (e.g., `torch.randn(size=(k,m)).cuda()*sigma+mu`) to significantly boost the computational efficiency for sampling from common distributions such as Gaussian. In Appendix H, we show that when using one NVIDIA A100 GPU, the runtime time of sampling Gaussian mixture $(1 - q)\mathcal{N}(0, \sigma^2) + q\mathcal{N}(1, \sigma^2)$ can be improved by $10^3$ times compared with CPU-only scenario.

**(3) Efficient *online privacy accounting*.** When training ML models with DP-SGD or its variants, a privacy practitioner usually wants to compute a running privacy leakage for *every* training iteration, and pick the checkpoint with the best utility-privacy tradeoff. This involves estimating $\delta_{Y^{(i)}}(\varepsilon)$ for every $i = 1, \ldots, k$, where $Y^{(i)} := \sum_{j=1}^i Y_j$. We refer to such a scenario as *online privacy accounting*[2]. MC accountant is especially efficient for online privacy accounting. When estimating

---

[2]Note that this is different from the scenario of privacy odometer [36], as here the privacy parameter of the next individual mechanism is not adaptively chosen.

$\delta_{Y^{(i)}}(\varepsilon)$, one can re-use the samples previously drawn from $Y_1, \ldots, Y_{i-1}$ that were used for estimating privacy loss at earlier iterations.

These advantages are justified empirically in Section 6 and Appendix H.

## 6 Numerical Experiments

In this section, we conduct numerical experiments to illustrate **(1)** EVR paradigm with MC verifiers enables a tighter privacy analysis, and **(2)** MC accountant achieves state-of-the-art performance in privacy parameter estimation.

### 6.1 EVR vs Upper Bound

To illustrate the advantage of the EVR paradigm compared with directly using a strict upper bound for privacy parameters, we take the current state-of-the-art DP accountant, the FFT-based method from [19] as the example.

**EVR provides a tighter privacy guarantee.**
Recall that in Figure 1, FFT-based method provides vacuous bound when the ground-truth $\delta_Y(\varepsilon) < 10^{-10}$. Under the same hyperparameter setting, Figure 3 (a) shows the privacy bound of the EVR paradigm where the $\delta^{\text{est}}$ are FFT's estimates. We use the Importance Sampling estimator $\widehat{\delta}_{\text{IS}}$ for DP verification. We experiment with different values of $\tau$. A higher value of $\tau$ leads to tighter privacy guarantee but longer runtime. For fair comparison, the EVR's output distribution needs to be almost indistinguishable from the original mechanism. We set $\rho = (1+\tau)/2$ and set $\Delta$ according to the heuristic from Theorem 14. This guarantees that, as long as the estimate of $\delta^{\text{est}}$ from FFT is not a big underestimation (i.e., as long as $\delta^{\text{est}} \geq \rho \delta_Y(\varepsilon)$), the failure probability of the EVR paradigm is negligible ($O(\delta_Y(\varepsilon))$). The 'FFT-EVR' curve in Figure 3 (a) is essentially the 'FFT-est' curve in Figure 1 scaled up by $1/\tau$. As we can see, EVR provides a significantly better privacy analysis in the regime where the 'FFT-upp' is unmeaningful ($\delta < 10^{-10}$).

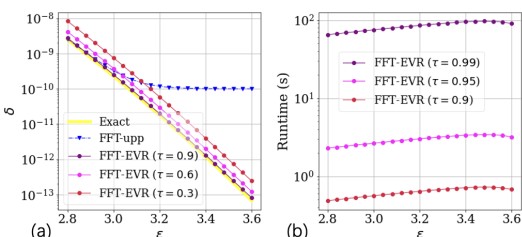

(a)

(b)

Figure 3: Privacy analysis and runtime of the EVR paradigm. The settings are the same as Figure 1. For (a), when $\tau > 0.9$, the curves are indistinguishable from 'Exact'. For fair comparison, we set $\rho = (1+\tau)/2$ and set $\Delta$ according to Theorem 14, which ensures EVR's failure probability of the order of $\delta$. For (b), the runtime is estimated on an NVIDIA A100-SXM4-80GB GPU.

**EVR incurs little extra runtime.** In Figure 3 (b), we plot the runtime of the Importance Sampling verifier in Figure 3 (b) for different $\tau \geq 0.9$. Note that for $\tau > 0.9$, the privacy curves are indistinguishable from 'Exact' in Figure 3 (a). The runtime of EVR is determined by the number of samples required to achieve the target $\tau$-relaxed FP rate from Theorem 13. Smaller $\tau$ leads to faster DP verification. As we can see, even when $\tau = 0.99$, the runtime of DP verification in the EVR is $< 2$ minutes. This is attributable to the sample-efficient IS estimator and GPU acceleration.

**EVR provides better privacy-utility tradeoff for Privacy-preserving ML with minimal time consumption.** To further underscore the superiority of the EVR paradigm in practical applications, we illustrate the privacy-utility tradeoff curve when finetuning on CIFAR100 dataset with DP-SGD. As shown in Figure 4, the EVR paradigm provides a lower test error across all privacy budget $\varepsilon$ compared with the traditional upper bound method. For instance, it achieves around 7% (relative) error reduction when $\varepsilon = 0.6$. The runtime time required for privacy verification is less than $< 10^{-10}$ seconds for all $\varepsilon$, which is negligible compared to the training time. We provide additional experimental results in Appendix H.

### 6.2 MC Accountant

We evaluate the MC Accountant proposed in Section 5. We focus on privacy accounting for the composition of Poisson Subsampled Gaussian mechanisms, the algorithm behind the famous DP-SGD algorithm [1]. The mechanism is specified by the noise magnitude $\sigma$ and subsampling rate $q$.

**Settings.** We consider two practical scenarios of privacy accounting: **(1) Offline accounting** which aims at estimating $\delta_{Y^{(k)}}(\varepsilon)$, and **(2) Online accounting** which aims at estimating $\delta_{Y^{(i)}}(\varepsilon)$ for all $i = 1, \ldots, k$. For space constraint, we only show the results of online accounting here, and defer the results for offline accounting to Appendix H. **Metric: Relative Error.** To easily and fairly evaluate the performance of privacy parameter estimation, we compute the *almost exact* (yet computationally expensive) privacy parameters as the ground-truth value. The ground-truth value allows us to compute the *relative error* of an estimate of privacy leakage. That is, if the corresponding ground-truth of an estimate $\widehat{\delta}$ is $\delta$, then the relative error $r_{\mathrm{err}} = |\widehat{\delta} - \delta|/\delta$. **Implementation.** For MC accountant, we use the IS estimator described in Section 4.2. For baselines, in addition to the FFT-based and CLT-based method we mentioned earlier, we also examine AFA [44] and GDP accountant [7]. For a fair comparison, we adjust the number of samples for MC accountant so that the runtime of MC accountant and FFT is comparable. Note that we compared with the privacy parameter *estimates* instead of upper

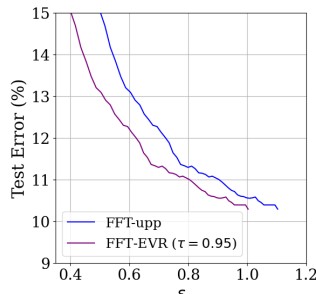

Figure 4: Utility-privacy tradeoff curve for fine-tuning ImageNet-pretrained BEiT [4] on CIFAR100 when $\delta = 10^{-5}$. We follow the training procedure from [34].

bounds from the baselines. Detailed settings for both MC accountant and the baselines are provided in Appendix H.

**Results for Online Accounting: MC accountant is both more accurate and efficient.** Figure 5 (a) shows the online accounting results for $(\sigma, \delta, q) = (1.0, 10^{-9}, 10^{-3})$. As we can see, MC accountant outperforms all of the baselines in estimating $\varepsilon_Y(\delta)$. The sharp decrease in FFT at approximately 250 steps is due to the transition of FFT's estimates from underestimating before this point to overestimating after. Figure 5 (b) shows that MC accountant is around 5 times faster than FFT, the baseline with the best performance in (a). This showcases the MC accountant's efficiency and accuracy in online setting.

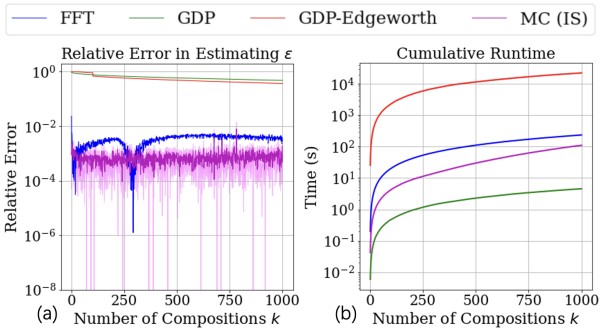

Figure 5: Experiment for Composing Subsampled Gaussian Mechanisms in the Online Setting. (a) Compares the relative error in approximating $k \mapsto \varepsilon_Y(\delta)$. The error bar for MC accountant is the variance taken over 5 independent runs. Note that the y-axis is in the log scale. (b) Compares the cumulative runtime for online privacy accounting. We did not show AFA [44] as it does not terminate in 24 hours.

## 7    Conclusion & Limitations

This paper tackles the challenge of deriving provable privacy leakage upper bounds in privacy accounting. We present the estimate-verify-release (EVR) paradigm which enables the safe use of privacy parameter estimate. **Limitations.** Currently, our MC-based DP verifier and accountant require known and efficiently samplable dominating pairs and PRV for the individual mechanism. Fortunately, this applies to commonly used mechanisms such as Gaussian mechanism and DP-SGD. Generalizing MC-based DP verifier and accountant to other mechanisms is an interesting future work.

## Acknowledgments

This work was supported in part by the National Science Foundation under grants CNS-2131938, CNS-1553437, CNS-1704105, the ARL's Army Artificial Intelligence Innovation Institute (A2I2), the Office of Naval Research Young Investigator Award, the Army Research Office Young Investigator Prize, Schmidt DataX award, and Princeton E-ffiliates Award, Amazon-Virginia Tech Initiative in Efficient and Robust Machine Learning, and Princeton's Gordon Y. S. Wu Fellowship. We are grateful to anonymous reviewers at NeurIPS for their valuable feedback.

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

# A  Extended Related Work

In this section, we review the related works in privacy accounting, privacy verification, privacy auditing, and we also discuss the connection between our EVR paradigm and the famous Propose-Test-Release (PTR) paradigm.

**Privacy Accounting.**    Early privacy accounting techniques such as Advanced Composition Theorem [15] only make use of the privacy parameters of the individual mechanisms, which bounds $\delta_{\mathcal{M}}(\varepsilon)$ in terms of the privacy parameter $(\varepsilon_i, \delta_i)$ for each $\mathcal{M}_i, i = 1, \ldots, k$. The optimal bound for $\delta_{\mathcal{M}}(\varepsilon)$ under this condition has been derived [22, 31]. However, the computation of the optimal bound is #P-hard in general. Bounding $\delta_{\mathcal{M}}(\varepsilon)$ only in terms of $(\varepsilon_i, \delta_i)$ is often sub-optimal for many commonly used mechanisms [31]. This disadvantage has spurred many recent advances in privacy accounting by making use of more statistical information from the specific mechanisms to be composed [1, 30, 24, 7, 23, 25, 19, 44, 17, 11, 41, 2]. All of these works can be described as approximating the expectation in (2) when $Y = \sum_{i=1}^{k} Y_i$. For instance, the line of [24, 7, 23, 25, 19, 17, 11] discretize the domain of each $Y_i$ and use Fast Fourier Transform (FFT) in order to speed up the approximation of $\delta_Y(\varepsilon)$. [44] tracks the characteristic function of the privacy loss random variables for the composed mechanism and still requires discretization when the mechanisms do not have closed-form characteristic functions. The line of [7, 41] uses Central Limit Theorem (CLT) to approximate the distribution of $Y = \sum_{i=1}^{k} Y_i$ as Gaussian distribution and uses the finite-sample bound to derive the strict upper bound for $\delta_Y(\varepsilon)$. We also note that [29] also uses Monte Carlo approaches to calculate optimal membership inference bounds. They use a similar Simple MC estimator as the one in Section 4.2. Although their Monte Carlo approach is similar, their error analysis only works for large values of $\delta$ ($\delta \approx 0.5$) as they use sub-optimal concentration bounds.

**Privacy Verification.**    As we mentioned in Remark 11, some previous works have also studied the problem of privacy verification. Most of the works consider either "white-box setting" where the input description of the DP mechanism is its corresponding randomized Boolean circuits [16, 8]. Some other works consider an even more stringent "black-box setting" where the verifier can only query the mechanism [18, 26, 6, 20, 28]. In contrast, our MC verifier is designed specifically for those mechanisms where the PRV can be derived, which includes many commonly used mechanisms such as the Subsampled Gaussian mechanism.

**Privacy verification via auditing.**    Several heuristics have tried to perform DP verification, forming a line of work called *auditing differential privacy* [21, 33, 27, 32, 38]. Specifically, these techniques can verify a claimed privacy parameter by computing a lower bound for the actual privacy parameter, and comparing that with the claimed privacy parameter. The input description of mechanism $\mathcal{M}$ for DPV, in this case, is a black-box oracle $\mathcal{M}(\cdot)$, where the DPV makes multiple queries to $\mathcal{M}(\cdot)$ and estimates the actual privacy leakage. Privacy auditing techniques can achieve 100% accuracy when $\delta^{\text{est}} > \delta_Y(\varepsilon)$ (or 0 $\rho$-FN rate for any $\rho \leq 1$), as the computed lower bound is guaranteed to be smaller than $\delta^{\text{est}}$. However, when $\delta^{\text{est}}$ lies between $\delta_Y(\varepsilon)$ and the computed lower bound, the DP verification will be wrong. Moreover, such techniques do not have a guarantee for the lower bound's tightness.

**Remark 15** (Connection between our EVR paradigm and the Propose-Test-Release (PTR) paradigm [12]). *PTR is a classic differential privacy paradigm introduced over a decade ago by [12], and is being generalized in [35, 42]. At a high level, PTR checks if releasing the query answer is safe with a certain amount of randomness (in a private way). If the test is passed, the query answer is released; otherwise, the program is terminated. PTR shares a similar underlying philosophy with our EVR paradigm. However, they are fundamentally different in terms of implementation. The verification step in EVR is completely independent of the dataset. In contrast, the test step in PTR measures the privacy risks for the mechanism $\mathcal{M}$ on a specific dataset $D$, which means that the test itself may cause additional privacy leakage. One way to think about the difference is that EVR asks "whether $\mathcal{M}$ is private", while PTR asks "whether $\mathcal{M}(D)$ is private".*

# B Proofs for Privacy

**Theorem 9.** *Algorithm 1 is $(\varepsilon, \delta^{\mathrm{est}}/\tau)$-DP for any $\tau > 0$ if $\mathsf{FP}_{DPV}(\varepsilon, \delta^{\mathrm{est}}; \tau) \le \delta^{\mathrm{est}}/\tau$.*

*Proof.* For any mechanism $\mathcal{M}$, we denote $A$ as the event that $\delta^{\mathrm{est}} \ge \tau \delta_Y(\varepsilon)$, and indicator variable $B = \mathbb{1}[\mathtt{DPV}(\mathcal{M}, \varepsilon, \delta^{\mathrm{est}}; \tau) = \mathtt{True}]$. Note that event $A$ implies $\mathcal{M}$ is $(\delta^{\mathrm{est}}/\tau)$-DP.

Thus, we know that

$$\Pr[B = 1|\bar{A}] \le \mathsf{FP}_{\mathsf{DPV}}(\varepsilon, \delta^{\mathrm{est}}; \tau) \tag{1}$$

For notation simplicity, we also denote $p_{\mathsf{FP}} := \Pr[B = 1|\bar{A}]$, and $p_{\mathsf{TP}} := \Pr[B = 1|A]$.

For any possible event $S$,

$$\Pr_{\mathcal{M}^{\mathrm{aug}}}[\mathcal{M}^{\mathrm{aug}}(D) \in S]$$
$$= \Pr_{\mathcal{M}}[\mathcal{M}(D) \in S|B = 1]\Pr[B = 1] + I[\bot \in S]\Pr[B = 0]$$
$$= \Pr_{\mathcal{M}}[\mathcal{M}(D) \in S|B = 1, A]\Pr[B = 1|A]I[A] + \Pr_{\mathcal{M}}[\mathcal{M}(D) \in S|B = 1, \bar{A}]\Pr[B = 1|\bar{A}]I[\bar{A}]$$
$$\quad + I[\bot \in S]\Pr[B = 0]$$
$$\le \left(e^{\varepsilon}\Pr_{\mathcal{M}}[\mathcal{M}(D') \in S|B = 1, A] + \frac{\delta^{\mathrm{est}}}{\tau}\right)\Pr[B = 1|A]I[A]$$
$$\quad + \Pr_{\mathcal{M}}[\mathcal{M}(D) \in S|B = 1, \bar{A}]\Pr[B = 1|\bar{A}]I[\bar{A}]$$
$$\quad + I[\bot \in S]\Pr[B = 0]$$
$$\le \left(e^{\varepsilon}\Pr_{\mathcal{M}}[\mathcal{M}(D') \in S|B = 1, A] + \frac{\delta^{\mathrm{est}}}{\tau}\right)p_{\mathsf{TP}}I[A] + p_{\mathsf{FP}}I[\bar{A}] + I[\bot \in S]\Pr[B = 0]$$
$$\le e^{\varepsilon}\left(\Pr_{\mathcal{M}}[\mathcal{M}(D') \in S|B = 1, A]p_{\mathsf{TP}}I[A] + \Pr_{\mathcal{M}}[\mathcal{M}(D') \in S|B = 1, \bar{A}]p_{\mathsf{FP}}I[\bar{A}] + I[\bot \in S]\Pr[B = 0]\right)$$
$$\quad + \frac{\delta^{\mathrm{est}}}{\tau}p_{\mathsf{TP}}I[A] + p_{\mathsf{FP}}I[\bar{A}]$$
$$\le e^{\varepsilon}\Pr_{\mathcal{M}^{\mathrm{aug}}}[\mathcal{M}^{\mathrm{aug}}(D') \in S] + \max\left(\frac{\delta^{\mathrm{est}}p_{\mathsf{TP}}}{\tau}, p_{\mathsf{FP}}\right)$$

where in the first inequality, we use the definition of differential privacy. Therefore, $\mathcal{M}^{\mathrm{aug}}$ is $\left(\varepsilon, \max\left(\frac{\delta^{\mathrm{est}}p_{\mathsf{TP}}}{\tau}, p_{\mathsf{FP}}\right)\right)$-DP. By assumption of $p_{\mathsf{FP}} \le \mathsf{FP}_{\mathsf{DPV}}(\varepsilon, \delta^{\mathrm{est}}; \tau) \le \delta^{\mathrm{est}}/\tau$, we reach the conclusion. $\qquad\square$

# C Importance Sampling via Exponential Tilting

**Notation Review.** Recall that most of the recently proposed DP accountants are essentially different techniques for estimating the expectation

$$\delta_{Y=\sum_{i=1}^{k} Y_i}(\varepsilon) = \mathbb{E}_Y \left[ \left(1 - e^{\varepsilon - Y}\right)_+ \right]$$

where each $Y_i$ is the privacy loss random variable $Y_i = \log\left(\frac{P_i(t)}{Q_i(t)}\right)$ for $t \sim P_i$, and $(P_i, Q_i)$ is a pair of dominating distribution for individual mechanism $\mathcal{M}_i$. In the following text, we denote the product distribution $\boldsymbol{P} := P_1 \times \ldots \times P_k$ and $\boldsymbol{Q} := Q_1 \times \ldots \times Q_k$. Recall from Lemma 5 that $(\boldsymbol{P}, \boldsymbol{Q})$ is a pair of dominating distributions for the composed mechanism $\mathcal{M}$. For notation simplicity, we denote a vector $\boldsymbol{t} := (t^{(1)}, \ldots, t^{(k)})$. We slightly abuse the notation and write $y(t; P, Q) := \log\left(\frac{P(t)}{Q(t)}\right)$. Note that $y(\boldsymbol{t}; \boldsymbol{P}, \boldsymbol{Q}) = \sum_{i=1}^{k} y(t^{(i)}; P_i, Q_i)$. When the context is clear, we omit the dominating pairs and simply write $y(t)$.

**Dominating Distribution Pairs for Poisson Subsampled Gaussian Mechanisms.** The dominating distribution pair for Poisson Subsampled Gaussian Mechanisms is a well-known result.

**Lemma 16.** *For Poisson Subsampled Gaussian mechanism with sensitivity $C$, noise variance $C^2\sigma^2$, and subsampling rate $q$, one dominating pair $(P, Q)$ is $Q := \mathcal{N}(0, \sigma^2)$ and $P := (1-q)\mathcal{N}(0, \sigma^2) + q\mathcal{N}(1, \sigma^2)$.*

*Proof.* See Appendix B of [19]. □

That is, $Q$ is just a 1-dimensional standard Gaussian distribution, and $P$ is a convex combination between standard Gaussian and a Gaussian centered at $1$.

**Remark 17** (Dominating pair supported on higher dimensional space)**.** *The cost of our approach would not increase (in terms of the number of samples) even if the dominating pair is supported in a high dimensional space. For Monte Carlo estimate, we can see from Hoeffding's inequality that the expected error rate of estimation is independent of the dimension of the support set of dominating distribution pairs. This means the number of samples we need to ensure a certain confidence interval is independent of the dimension. However, we should also note that although the number of samples does not change, the sampling process itself might be more costly for higher dimensional spaces.*

## C.1 Importance Sampling for the Composition of Poisson Subsampled Gaussian Mechanisms

Importance Sampling (IS) is a classic method for rare event simulation. It samples from an alternative distribution instead of the distribution of the quantity of interest, and a weighting factor is then used for correcting the difference between the two distributions. Specifically, we can re-write the expression for $\delta_Y(\varepsilon)$ as follows:

$$
\begin{aligned}
\delta_Y(\varepsilon) &= \mathbb{E}_Y\left[(1 - e^{\varepsilon - Y})_+\right] \\
&= \mathbb{E}_{\boldsymbol{t} \sim \boldsymbol{P}}\left[\left(1 - e^{\varepsilon - y(\boldsymbol{t}; \boldsymbol{P}, \boldsymbol{Q})}\right)_+\right] \\
&= \mathbb{E}_{\boldsymbol{t} \sim \boldsymbol{P}'}\left[\left(1 - e^{\varepsilon - y(\boldsymbol{t}; \boldsymbol{P}, \boldsymbol{Q})}\right)_+ \frac{\boldsymbol{P}(\boldsymbol{t})}{\boldsymbol{P}'(\boldsymbol{t})}\right]
\end{aligned}
\tag{2}
$$

where $\boldsymbol{P}'$ is the alternative distribution up to the user's choice. From Equation (2), one can construct an unbiased importance sampling estimator for $\delta_Y(\varepsilon)$ by sampling from $\boldsymbol{P}'$. In this section, we develop a $\boldsymbol{P}'$ for estimating $\delta_Y(\varepsilon)$ when composing *identically distributed Poisson subsampled Gaussian mechanisms*, which is arguably the most important DP mechanism nowadays due to its application in differentially private stochastic gradient descent.

*Exponential tilting* is a common way to construct alternative sampling distribution for IS. The exponential tilting of a distribution $P$ is defined as

$$P_\theta(t) := \frac{e^{\theta t}}{M_P(\theta)} P(t)$$

where $M_P(\theta) := \mathbb{E}_{t \sim P}[e^{\theta t}]$ is the moment generating function for $P$. Such a transformation is especially convenient for distributions from the exponential family. For example, for normal distribution $\mathcal{N}(\mu, \sigma^2)$, the tilted distribution is $\mathcal{N}(\mu + \theta\sigma^2, \sigma^2)$, which is easy to sample from.

Without the loss of generality, we consider Poisson Subsampled Gaussian mechanism with sensitivity 1, noise variance $\sigma^2$, and subsampling rate $q$. Recall from Lemma 16 that the dominating pair in this case is $Q := \mathcal{N}(0, \sigma^2)$ and $P := (1-q)\mathcal{N}(0, \sigma^2) + q\mathcal{N}(1, \sigma^2)$. For notation simplicity, we denote $P_0 := \mathcal{N}(1, \sigma^2)$, and thus $P = (1-q)Q + qP_0$. Since each individual mechanism is the same, $\boldsymbol{P} = P \times \ldots \times P$ and $\boldsymbol{Q} = Q \times \ldots \times Q$. The exponential tilting of $P$ with parameter $\theta$ is $P_\theta := (1-q)\mathcal{N}(\theta\sigma^2, \sigma^2) + q\mathcal{N}(1 + \theta\sigma^2, \sigma^2)$. We propose the following importance sampling estimator for $\delta_Y(\varepsilon)$ based on exponential tilting.

**Definition 18** (Importance Sampling Estimator for Subsampled Gaussian Composition). *Let the alternative distribution*

$$\boldsymbol{P}' := \boldsymbol{P}_\theta = (P, \ldots, \underbrace{P_\theta}_{i\text{th dim}}, \ldots, P), \quad i \sim \mathrm{Unif}([k])$$

*with $\theta = 1/2 + \sigma^2 \log\left(\frac{\exp(\varepsilon) - (1-q)}{q}\right)$. Given a random draw $\boldsymbol{t} \sim \boldsymbol{P}_\theta$, an unbiased sample for $\delta_Y(\varepsilon)$ is $\left(1 - e^{\varepsilon - y(\boldsymbol{t}; \boldsymbol{P}, \boldsymbol{Q})}\right)_+ \left(\frac{1}{k}\sum_{i=1}^{k} \frac{P_\theta(t_i)}{P(t_i)}\right)^{-1}$. We denote $\widehat{\boldsymbol{\delta}}_{\mathrm{IS},\theta}^m(\varepsilon)$ as the random variable of the corresponding importance sampling estimator with $m$ samples.*

We defer the formal justification of the choice of $\theta$ to Appendix C.2. We first give the intuition for why we choose such an alternative distribution $\boldsymbol{P}_\theta$.

**Intuition for the alternative distribution $\boldsymbol{P}_\theta$.** It is well-known that the variance of the importance sampling estimator is minimized when the alternative distribution

$$\boldsymbol{P}'(\boldsymbol{t}) \propto \left(1 - e^{\varepsilon - y(\boldsymbol{t})}\right)_+ \boldsymbol{P}(\boldsymbol{t})$$

The distribution of each privacy loss random variable $y(t; P, Q), t \sim P$ is light-tailed, which means that for the rare event where $y(\boldsymbol{t}) = \sum_{i=1}^{k} y(t^{(i)}) > \varepsilon$, it is most likely that there is only *one* outlier $t^*$ among all $\{t^{(i)}\}_{i=1}^{k}$ such that $y(t^*)$ is large (which means that $y(\boldsymbol{t})$ is also large), and all the rest of $y(t^{(i)})$s are small. Hence, a reasonable alternative distribution can just tilt the distribution of a randomly picked $t^{(i)}$, and leave the rest of $k-1$ distributions to stay the same. Moreover, $\theta$ is selected to *approximately* minimize the variance of $\widehat{\boldsymbol{\delta}}_{\mathrm{IS},\theta}$ (detailed in Appendix C.2). An intuitive way to see it is that $y(\theta) = \varepsilon$, which significantly improves the probability where $y(\boldsymbol{t}) \geq \varepsilon$ while also accounting for the fact that $P(t)$ decays exponentially fast as $t$ increases.

We also empirically verify the advantage of the IS estimator over the SMC estimator. The orange curve in Figure 6 (a) shows the empirical estimate of $\mathbb{E}[(\widehat{\boldsymbol{\delta}}_{\mathrm{IS},\theta})^2]$ which quantifies the variance of $\widehat{\boldsymbol{\delta}}_{\mathrm{IS},\theta}$. Note that $\theta = 0$ corresponds to the case of $\widehat{\boldsymbol{\delta}}_{\mathrm{MC}}$. As we can see, $\mathbb{E}[(\widehat{\boldsymbol{\delta}}_{\mathrm{IS},\theta})^2]$ drops quickly as $\theta$ increases, and eventually converges. We can also see that the $\theta$ selected by our heuristic in Definition 18 (marked as red '*') approximately corresponds to the lowest point of $\mathbb{E}[(\widehat{\boldsymbol{\delta}}_{\mathrm{IS},\theta})^2]$. This validates our theoretical justification for the selection of $\theta$.

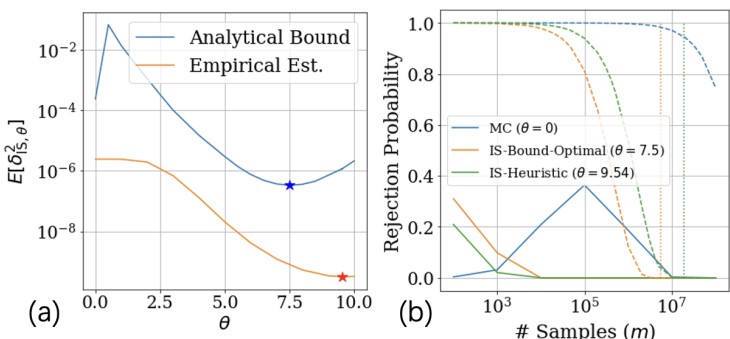

(a)

(b)

Figure 6: We examine the properties of MC-based DP verifiers for Poisson Subampled mechanism. We set $q = 10^{-3}, \sigma = 0.6, \varepsilon = 1.5, k = 100$. $\delta_Y(\varepsilon) \approx 7.7 \times 10^{-6}$ in this case. **(a)** Plot for the upper bound and empirical estimate of $\mathbb{E}[\widehat{\delta}_{\text{MC}}^2]$ and $\mathbb{E}[\widehat{\delta}_{\text{IS},\theta}^2]$. The upper bounds are computed by Corollary 22 (for $\mathbb{E}[\widehat{\delta}_{\text{MC}}^2]$) and Theorem 23 (for $\mathbb{E}[\widehat{\delta}_{\text{IS},\theta}^2]$). Note that $\theta = 0$ corresponds to $\widehat{\delta}_{\text{MC}}$. The red star indicates the second moment for the value of $\theta$ selected by our heuristic in Definition 18. The blue star indicates the $\theta$ that minimizes the analytical bound. **(b)** Empirical estimate of the rejection probability $\Pr[\widehat{\delta}_{\text{IS},\theta}^m > \delta^{\text{est}}/\tau - \Delta]$ scaled with the number of samples $m$. We set $\delta^{\text{est}} = 0.8\delta_Y(\varepsilon), \tau = 10^{-5}/\delta^{\text{est}}$, and we set $\Delta$ following the heuristic proposed in Section 4.4.

## C.2 Justification of the Heuristic of Choosing Exponential Tilting Parameter $\theta$

The variance of the IS estimator proposed in Definition 18 is given by

$$
\mathbb{E}_{\boldsymbol{t} \sim \boldsymbol{P}_\theta} \left[ \left(1 - e^{\varepsilon - y(\boldsymbol{t};\boldsymbol{P},\boldsymbol{Q})}\right)_+^2 \left(\frac{\boldsymbol{P}(\boldsymbol{t})}{\boldsymbol{P}_\theta(\boldsymbol{t})}\right)^2 \right]
$$

$$
= \mathbb{E}_{\boldsymbol{t} \sim \boldsymbol{P}} \left[ \left(1 - e^{\varepsilon - y(\boldsymbol{t};\boldsymbol{P},\boldsymbol{Q})}\right)_+^2 \left(\frac{\boldsymbol{P}(\boldsymbol{t})}{\boldsymbol{P}_\theta(\boldsymbol{t})}\right) \right]
$$

$$
= \mathbb{E}_{\boldsymbol{t} \sim \boldsymbol{P}} \left[ \left(1 - e^{\varepsilon - y(\boldsymbol{t};\boldsymbol{P},\boldsymbol{Q})}\right)_+^2 \left(\frac{1}{k}\sum_{i=1}^k \frac{P_\theta(t_i)}{P(t_i)}\right)^{-1} \right]
$$

$$
= k M_P(\theta) \mathbb{E}_{\boldsymbol{t} \sim \boldsymbol{P}} \left[ \left(1 - e^{\varepsilon - y(\boldsymbol{t};\boldsymbol{P},\boldsymbol{Q})}\right)_+^2 \left(\frac{1}{\sum_{i=1}^k e^{\theta t_i}}\right) \right]
$$

Let $S(\theta) := k M_P(\theta) \mathbb{E}_{\boldsymbol{t} \sim \boldsymbol{P}} \left[ \left(1 - e^{\varepsilon - y(\boldsymbol{t};\boldsymbol{P},\boldsymbol{Q})}\right)_+^2 \left(\frac{1}{\sum_{i=1}^k e^{\theta t_i}}\right) \right]$. We aim to find $\theta$ that minimizes $S(\theta)$.

Note that

$$
M_P(\theta) = (1-q)e^{\frac{1}{2}\sigma^2\theta^2} + qe^{\theta + \frac{1}{2}\sigma^2\theta^2} \tag{3}
$$

To simplify the notation, let $b(\boldsymbol{t}) := \left(1 - e^{\varepsilon - y(\boldsymbol{t})}\right)_+^2 \prod_{i=1}^k P_0(t_i)$.

$$
\frac{\partial}{\partial \theta} S(\theta) = \left[(1-q)e^{\frac{1}{2}\sigma^2\theta^2}(\sigma^2\theta) + qe^{\theta+\frac{1}{2}\sigma^2\theta^2}(1+\sigma^2\theta)\right] \int \cdots \int b(\boldsymbol{t}) \left(\sum_{i=1}^k e^{\theta t_i}\right)^{-1} d\boldsymbol{t} \tag{4}
$$

$$
- \left[(1-q)e^{\frac{1}{2}\sigma^2\theta^2} + qe^{\theta+\frac{1}{2}\sigma^2\theta^2}\right] \int \cdots \int b(\boldsymbol{t}) \frac{\sum_{i=1}^k e^{\theta t_i} t_i}{\left(\sum_{i=1}^k e^{\theta t_i}\right)^2} d\boldsymbol{t} \tag{5}
$$

By setting $\frac{\partial}{\partial \theta} S(\theta) = 0$ and simplify the expression, we have

$$\frac{(1 - q + qe^\theta)(\sigma^2\theta) + qe^\theta}{1 - q + qe^\theta} = \frac{\int \ldots \int b(\boldsymbol{t}) \frac{\sum_{i=1}^k e^{\theta t_i} t_i}{\left(\sum_{i=1}^k e^{\theta t_i}\right)^2} d\boldsymbol{t}}{\int \ldots \int b(\boldsymbol{t}) \left(\sum_{i=1}^k e^{\theta t_i}\right)^{-1} d\boldsymbol{t}} \tag{6}$$

As we mentioned earlier, $b(\boldsymbol{t}) > 0$ only when $y(\boldsymbol{t}) = \sum_{i=1}^k y(t^{(i)}) > \varepsilon$, and for such an event it is most likely that there is only *one outlier* $t^*$ among all $\{t^{(i)}\}_{i=1}^k$ such that $y(t^*) \approx \varepsilon$, and all the rest of $y(t^{(i)}) \approx 0$. Therefore, a simple but surprisingly effective approximation for the RHS of (6) is

$$\frac{\int \ldots \int b(\boldsymbol{t}) \frac{\sum_{i=1}^k e^{\theta t_i} t_i}{\left(\sum_{i=1}^k e^{\theta t_i}\right)^2} d\boldsymbol{t}}{\int \ldots \int b(\boldsymbol{t}) \left(\sum_{i=1}^k e^{\theta t_i}\right)^{-1} d\boldsymbol{t}} \approx \frac{b(t)e^{-\theta t} t}{b(t)e^{-\theta t}} = t \tag{7}$$

for $t$ s.t. $y(t) = \varepsilon$. This leads to an approximate solution

$$\theta^* = \frac{1}{2\sigma^2} + \log\left((\exp(\varepsilon) - (1 - q))/q\right) \tag{8}$$

# D    Sample Complexity for Achieving Target False Positive Rate

To derive the sample complexity for achieving a DP verifier with a target false positive rate, we use Bennett's inequality.

**Lemma 19** (Bennett's inequality)**.** *Let $X_1, \ldots, X_n$ be independent real-valued random variables with finite variance such that $X_i \leq b$ for some $b > 0$ almost surely for all $1 \leq i \leq n$. Let $\nu \geq \sum_{i=1}^n \mathbb{E}[X_i^2]$. For any $t > 0$, we have*

$$\Pr\left[ \sum_{i=1}^n X_i - \mathbb{E}[X_i] \geq t \right] \leq \exp\left( -\frac{\nu}{b^2} h\left( \frac{bt}{\nu} \right) \right) \tag{9}$$

*where $h(x) = (1 + x)\log(1 + x) - x$ for $x > 0$.*

**Theorem 13.** *Suppose $\mathbb{E}\left[ \left( \widehat{\boldsymbol{\delta}}_{MC} \right)^2 \right] \leq \nu$. DPV instantiated by $\widehat{\boldsymbol{\delta}}_{MC}^m$ has bounded $\tau$-relaxed false positive rate $\mathsf{FP}_{MC}(\varepsilon, \delta^{\mathrm{est}}; \tau) \leq \delta^{\mathrm{est}}/\tau$ with $m \geq \frac{2\nu}{\Delta^2} \log(\tau/\delta^{\mathrm{est}})$.*

*Proof.* For any $\mathcal{M}$ s.t. $\delta^{\mathrm{est}} < \tau \delta_Y(\varepsilon)$, we have

$$\Pr\left[ \widehat{\boldsymbol{\delta}}_{\mathrm{MC}}^m(\varepsilon; Y) < \delta^{\mathrm{est}}/\tau - \Delta \right] \leq \Pr\left[ \widehat{\boldsymbol{\delta}}_{\mathrm{MC}}^m(\varepsilon; Y) < \delta_Y(\varepsilon) - \Delta \right]$$

$$= \Pr\left[ \frac{1}{m} \sum_{i=1}^m (\widehat{\boldsymbol{\delta}}_{\mathrm{MC}}^{(i)} - \delta_Y(\varepsilon)) < -\Delta \right]$$

$$= \Pr\left[ \sum_{i=1}^m (\delta_Y(\varepsilon) - \widehat{\boldsymbol{\delta}}_{\mathrm{MC}}^{(i)}) > m\Delta \right] \tag{10}$$

Since $\widehat{\boldsymbol{\delta}}_{\mathrm{MC}} \in [-1, 0]$, the condition in Bennett's inequality is satisfied with $b \to 0^+$. Hence, (10) can be upper bounded by

$$(10) \leq \lim_{b \to 0^+} \exp\left( -\frac{m\nu}{b^2} h\left( \frac{b\Delta}{\nu} \right) \right)$$

$$= \exp\left( -\frac{m\Delta^2}{2\nu} \right)$$

By setting $\exp\left( -\frac{m\Delta^2}{2\nu} \right) \leq \delta^{\mathrm{est}}/\tau$, we have

$$m \geq \frac{2\nu}{\Delta^2} \log(\tau/\delta^{\mathrm{est}})$$

$\square$

# E  Proofs for Moment Bound

## E.1  Overview

As suggested by Theorem 13, a good upper bound for $\mathbb{E}\left[\left(\widehat{\boldsymbol{\delta}}_{\mathtt{MC}}\right)^2\right]$ (or $\mathbb{E}\left[\left(\widehat{\boldsymbol{\delta}}_{\mathtt{IS}}\right)^2\right]$) is important for the computational efficiency of MC-based DPV.

We upper bound the higher moment of $\widehat{\boldsymbol{\delta}}_{\mathtt{MC}}$ through the RDP guarantee for the composed mechanism $\mathcal{M}$. This is a natural idea since converting RDP to upper bounds for $\delta_Y(\varepsilon)$ – the first moment of $\widehat{\boldsymbol{\delta}}_{\mathtt{MC}}$ – is a well-studied problem [30, 9, 3]. Recall that the RDP guarantee for $\mathcal{M}$ is equivalent to a bound for $M_Y(\lambda) := \mathbb{E}[e^{\lambda Y}]$ for $\mathcal{M}$'s privacy loss random variable $Y$ for any $\lambda \geq 0$.

**Lemma 20** (RDP-MGF bound conversion [30]). *If a mechanism $\mathcal{M}$ is $(\alpha, \varepsilon_{\mathrm{R}}(\alpha))$-RDP, then* $M_Y(\lambda) \leq \exp(\lambda \varepsilon_{\mathrm{R}}(\lambda + 1))$.

We convert an upper bound for $M_Y(\cdot)$ into the following guarantee for the higher moment of $\widehat{\boldsymbol{\delta}}_{\mathtt{MC}} = (1 - e^{\varepsilon - Y})_+$.

**Theorem 21.** *For any $u \geq 1$, we have*

$$\mathbb{E}[(\widehat{\boldsymbol{\delta}}_{\mathtt{MC}})^u] = \mathbb{E}\left[(1 - e^{\varepsilon - Y})_+^u\right] \leq \min_{\lambda \geq 0} M_Y(\lambda) e^{-\varepsilon \lambda} \frac{u^u \lambda^\lambda}{(u + \lambda)^{u + \lambda}}$$

The proof is deferred to Appendix E.2. The basic idea is to find the smallest constant $c$ such that $\mathbb{E}[(\widehat{\boldsymbol{\delta}}_{\mathtt{MC}})^u] \leq c M_Y(\lambda)$. By setting $u = 1$, our result recovers the RDP-DP conversion from [9]. By setting $u = 2$, we obtain the desired bound for $\mathbb{E}[(\widehat{\boldsymbol{\delta}}_{\mathtt{MC}})^2]$.

**Corollary 22.** $\mathbb{E}[(\widehat{\boldsymbol{\delta}}_{\mathtt{MC}})^2] \leq \min_{\lambda \geq 0} M_Y(\lambda) e^{-\varepsilon \lambda} \frac{4 \lambda^\lambda}{(\lambda + 2)^{\lambda + 2}}$.

Corollary 22 applies to any mechanisms where the RDP guarantee is available, which covers a wide range of commonly used mechanisms such as (Subsampled) Gaussian or Laplace mechanism. We also note that one may be able to further tighten the above bound similar to the optimal RDP-DP conversion in [3]. We leave this as an interesting future work.

Next, we derive the upper bound for $\mathbb{E}[(\widehat{\boldsymbol{\delta}}_{\mathtt{IS}, \theta})^2]$ for Poisson Subsampled Gaussian mechanism.

**Theorem 23.** *For any positive integer $\lambda$, and for any $a, b \geq 1$ s.t. $1/a + 1/b = 1$, we have* $\mathbb{E}[(\widehat{\boldsymbol{\delta}}_{IS, \theta})^2] \leq k M_P(\theta) \left(\mathbb{E}[\widehat{\boldsymbol{\delta}}_{MC}^{2a}]\right)^{1/a} \cdot \left(b \theta e^{-\lambda \varepsilon} \int [r(\lambda, x)]^k e^{-b \theta x} dx\right)^{1/b}$ *where $r(\lambda, x)$ is an upper bound for $\Pr_{\boldsymbol{t} \sim \boldsymbol{P}}[\max_i t_i \leq x, y(\boldsymbol{t}) \geq \varepsilon]$ detailed in Appendix E.3.*

The proof is based on applying Hölder's inequality to the expression of $\mathbb{E}[(\widehat{\boldsymbol{\delta}}_{\mathtt{IS}, \theta})^2]$, and then bound the part where $\theta$ is involved: $\mathbb{E}_{\boldsymbol{t} \sim \boldsymbol{P}}\left[\left(\frac{1}{k} \sum_{i=1}^k \frac{P_\theta(t_i)}{P(t_i)}\right)^{-1}\right]$. We can bound $\mathbb{E}[\widehat{\boldsymbol{\delta}}_{\mathtt{MC}}^{2a}]$ through Theorem 21.

Figure 6 (a) shows the analytical bound from Corollary 22 and Theorem 23 compared with empirically estimated $\mathbb{E}[(\widehat{\boldsymbol{\delta}}_{\mathtt{MC}})^2]$ and $\mathbb{E}[(\widehat{\boldsymbol{\delta}}_{\mathtt{IS}, \theta})^2]$. As we can see, the analytical bound for $\mathbb{E}[(\widehat{\boldsymbol{\delta}}_{\mathtt{IS}, \theta})^2]$ for relatively large $\theta$ is much smaller than the bound for $\mathbb{E}[(\widehat{\boldsymbol{\delta}}_{\mathtt{MC}})^2]$ (i.e., $\theta = 0$ in the plot). Moreover, we find that the $\theta$ which minimizes the analytical bound (the blue '*') is *close* to the $\theta$ selected by our heuristic (the red '*'). For computational efficiency, one may prefer to use $\theta$ that minimizes the analytical bound. However, the heuristically selected $\theta$ is still useful when one simply wants to estimate $\delta_Y(\varepsilon)$ and does not require the formal, analytical guarantee for the false positive rate. We see such a scenario when we introduce the MC accountant in Section 5. We also note that such a discrepancy (and the gap between the analytical bound and empirical estimate) is due to the use of Hölder's inequality in bounding $\mathbb{E}[(\widehat{\boldsymbol{\delta}}_{\mathtt{IS}, \theta})^2]$. Further tightening the bound for $\mathbb{E}[(\widehat{\boldsymbol{\delta}}_{\mathtt{IS}, \theta})^2]$ is important for future work.

## E.2  Moment Bound for Simple Monte Carlo Estimator

**Theorem 21.** *For any $u \geq 1$, we have*

$$\mathbb{E}[(\widehat{\boldsymbol{\delta}}_{\mathtt{MC}})^u] = \mathbb{E}\left[(1 - e^{\varepsilon - Y})_+^u\right] \leq \min_{\lambda \geq 0} M_Y(\lambda) e^{-\varepsilon \lambda} \frac{u^u \lambda^\lambda}{(u + \lambda)^{u + \lambda}}$$

*Proof.*

$$\mathbb{E}[(1 - e^{\varepsilon - Y})_+^u] = \int (1 - e^{\varepsilon - x})_+^u P(x) \mathrm{d}x$$

$$= M_Y(\lambda) \int (1 - e^{\varepsilon - x})_+^u e^{-\lambda x} \frac{P(x) e^{\lambda x}}{\mathbb{E}[e^{\lambda Y}]} \mathrm{d}x$$

$$= M_Y(\lambda) \mathbb{E}_{x \sim P_\theta}[(1 - e^{\varepsilon - x})_+^u e^{-\lambda x}]$$

$$= M_Y(\lambda) e^{-\lambda \varepsilon} \mathbb{E}_{x \sim P_\theta}[(1 - e^{\varepsilon - x})_+^u e^{(\varepsilon - x)\lambda}]$$

where $P_\lambda(x) := \frac{P(x) e^{\lambda x}}{\mathbb{E}[e^{\lambda Y}]}$ is the exponential tilting of $P$. Define $f(x, \lambda) := (1 - e^{-x})_+^u e^{-x\lambda}$. When $x \leq 0$, $f(x, \lambda) = 0$. When $x > 0$, the derivative of $f$ with respect to $x$ is

$$\frac{\partial f(x, \lambda)}{\partial x} = e^{-x\lambda}(1 - e^{-x})^{u-1}[e^{-x}(u + \lambda) - \lambda]$$

It is easy to see that the maximum of $f(x, \lambda)$ is achieved at $x^* = \log\left(\frac{u+\lambda}{\lambda}\right)$, and we have

$$\max_x f(x, \lambda) = \left(\frac{u}{u + \lambda}\right)^u \left(\frac{\lambda}{u + \lambda}\right)^\lambda$$

$$= \frac{u^u \lambda^\lambda}{(u + \lambda)^{u+\lambda}}$$

Overall, we have

$$\mathbb{E}[(1 - e^{\varepsilon - Y})_+^u] \leq M_Y(\lambda) e^{-\varepsilon\lambda} \frac{u^u \lambda^\lambda}{(u + \lambda)^{u+\lambda}}$$

$\square$

### E.3    Moment Bound for Importance Sampling Estimator

In this section, we first prove two possible upper bounds for $\mathbb{E}[(\widehat{\boldsymbol{\delta}}_{\text{IS},\theta})^2]$ in Theorem 24 and Theorem 26. We then combine these two bounds in Theorem 23 via Holder's inequality.

**Theorem 24.** *For any $\theta \geq 1/\sigma^2$, we have*

$$\mathbb{E}[(\widehat{\boldsymbol{\delta}}_{\text{IS},\theta})^2] \leq M_P(\theta) \left[\frac{1}{k}\left(\frac{\varepsilon}{q} + k\right)\right]^{-\theta\sigma^2} e^{-\theta/2} \mathbb{E}[(\widehat{\boldsymbol{\delta}}_{\text{MC}})^2]$$

*Proof.*

$$\mathbb{E}_{\boldsymbol{t} \sim \boldsymbol{P_\theta}}\left[\left(1 - e^{\varepsilon - y(\boldsymbol{t};\boldsymbol{P},\boldsymbol{Q})}\right)_+^2 \left(\frac{\boldsymbol{P(t)}}{\boldsymbol{P_\theta(t)}}\right)^2\right]$$

$$= \mathbb{E}_{\boldsymbol{t} \sim \boldsymbol{P}}\left[\left(1 - e^{\varepsilon - y(\boldsymbol{t};\boldsymbol{P},\boldsymbol{Q})}\right)_+^2 \left(\frac{\boldsymbol{P(t)}}{\boldsymbol{P_\theta(t)}}\right)\right]$$

$$= \mathbb{E}_{\boldsymbol{t} \sim \boldsymbol{P}}\left[\left(1 - e^{\varepsilon - y(\boldsymbol{t};\boldsymbol{P},\boldsymbol{Q})}\right)_+^2 \left(\frac{1}{k}\sum_{i=1}^k \frac{P_\theta(t_i)}{P(t_i)}\right)^{-1}\right]$$

$$= M_P(\theta) \mathbb{E}_{\boldsymbol{t} \sim \boldsymbol{P}}\left[\left(1 - e^{\varepsilon - y(\boldsymbol{t};\boldsymbol{P},\boldsymbol{Q})}\right)_+^2 \left(\frac{k}{\sum_{i=1}^k e^{\theta t_i}}\right)\right] \tag{11}$$

Note that

$$\mathbb{E}_{\boldsymbol{t} \sim \boldsymbol{P}}\left[\left(1 - e^{\varepsilon - y(\boldsymbol{t};\boldsymbol{P},\boldsymbol{Q})}\right)_+^2 \left(\frac{k}{\sum_{i=1}^k e^{\theta t_i}}\right)\right]$$

$$= \mathbb{E}_{\boldsymbol{t} \sim \boldsymbol{P}}\left[\left(1 - e^{\varepsilon - y(\boldsymbol{t};\boldsymbol{P},\boldsymbol{Q})}\right)_+^2 \left(\frac{k}{\sum_{i=1}^k e^{\theta t_i}}\right) I[y(\boldsymbol{t};\boldsymbol{P},\boldsymbol{Q}) \geq \varepsilon]\right]$$

**Lemma 25.** *When* $y(\boldsymbol{t}; \boldsymbol{P}, \boldsymbol{Q}) = \sum_{i=1}^{k} \log\left(1 - q + qe^{(2t_i - 1)/(2\sigma^2)}\right) \geq \varepsilon$ *and* $\theta\sigma^2 \geq 1$, *we have*

$$\sum_{i=1}^{k} e^{\theta t_i} \geq k \left[\frac{1}{k}\left(\frac{\varepsilon}{k} + k\right) e^{1/(2\sigma^2)}\right]^{\theta\sigma^2}$$

*Proof.* Since $\log(1 + x) \leq x$, we have

$$\varepsilon \leq \sum_{i=1}^{k} \log\left(1 - q + qe^{(2t_i - 1)/(2\sigma^2)}\right)$$

$$\leq \sum_{i=1}^{k} q\left(e^{(2t_i - 1)/(2\sigma^2)} - 1\right)$$

Hence,

$$\sum_{i=1}^{k} e^{t_i/\sigma^2} \geq \left(\frac{\varepsilon}{q} + k\right) e^{1/(2\sigma^2)}$$

Hence,

$$\sum_{i=1}^{k} e^{t_i\theta} = \sum_{i=1}^{k} \left(e^{t_i/\sigma^2}\right)^{\theta\sigma^2} = k\left[\frac{1}{k}\sum_{i=1}^{k}\left(e^{t_i/\sigma^2}\right)^{\theta\sigma^2}\right]$$

$$\geq k\left[\frac{1}{k}\sum_{i=1}^{k}\left(e^{t_i/\sigma^2}\right)\right]^{\theta\sigma^2}$$

$$\geq k\left[\frac{1}{k}\left(\frac{\varepsilon}{q} + k\right)e^{1/(2\sigma^2)}\right]^{\theta\sigma^2}$$

$$= k\left[\frac{1}{k}\left(\frac{\varepsilon}{q} + k\right)\right]^{\theta\sigma^2} e^{\theta/2}$$

where the first inequality is due to Jensen's inequality. $\qquad\square$

By Lemma 25, we have

$$(11) \leq M_P(\theta)\frac{1}{\left[\frac{1}{k}\left(\frac{\varepsilon}{q} + k\right)\right]^{\theta\sigma^2} e^{\theta/2}}\mathbb{E}_{\boldsymbol{t}\sim\boldsymbol{P}}\left[\left(1 - e^{\varepsilon - y(\boldsymbol{t};\boldsymbol{P},\boldsymbol{Q})}\right)_+^2\right]$$

$$= M_P(\theta)\left[\frac{1}{k}\left(\frac{\varepsilon}{q} + k\right)\right]^{-\theta\sigma^2} e^{-\theta/2}\mathbb{E}[(\widehat{\boldsymbol{\delta}}_{\text{MC}})^2]$$

which concludes the proof. $\qquad\square$

**Theorem 26.** *For any positive integer $\lambda$, we have*

$$\mathbb{E}[(\widehat{\boldsymbol{\delta}}_{IS,\theta})^2] \leq kM_P(\theta)\theta e^{-\lambda\varepsilon}\int [r(\lambda, x)]^k e^{-\theta x}dx \tag{12}$$

*where $r(\lambda, x)$ is an upper bound for the MGF of privacy loss random variable of* truncated *Gaussian mixture $P|_{\leq x}$.*

*Proof.* Similar to Theorem 24, the goal is to bound

$$kM_P(\theta)\mathbb{E}_{\boldsymbol{t}\sim\boldsymbol{P}}\left[\left(1 - e^{\varepsilon - y(\boldsymbol{t};\boldsymbol{P},\boldsymbol{Q})}\right)_+^2\left(\frac{1}{\sum_{i=1}^{k} e^{\theta t_i}}\right)\right]$$

Note that

$$\mathbb{E}_{\boldsymbol{t}\sim\boldsymbol{P}}\left[\left(1-e^{\varepsilon-y(\boldsymbol{t};\boldsymbol{P},\boldsymbol{Q})}\right)_+^2\left(\frac{1}{\sum_{i=1}^k e^{\theta t_i}}\right)\right]$$

$$= \mathbb{E}_{\boldsymbol{t}\sim\boldsymbol{P}}\left[\left(1-e^{\varepsilon-y(\boldsymbol{t};\boldsymbol{P},\boldsymbol{Q})}\right)_+^2\left(\frac{1}{\sum_{i=1}^k e^{\theta t_i}}\right)I[y(\boldsymbol{t};\boldsymbol{P},\boldsymbol{Q})\geq\varepsilon]\right]$$

$$\leq \mathbb{E}_{\boldsymbol{t}\sim\boldsymbol{P}}\left[\frac{1}{\sum_{i=1}^k e^{\theta t_i}}I[y(\boldsymbol{t};\boldsymbol{P},\boldsymbol{Q})\geq\varepsilon]\right]$$

$$\leq \mathbb{E}_{\boldsymbol{t}\sim\boldsymbol{P}}\left[\frac{1}{e^{\theta t_{\max}}}I[y(\boldsymbol{t};\boldsymbol{P},\boldsymbol{Q})\geq\varepsilon]\right] \tag{13}$$

where $t_{\max}:=\max_i t_i$.

Further note that

$$(13) = \mathbb{E}_{\boldsymbol{t}\sim\boldsymbol{P}}\left[e^{-\theta t_{\max}}I[y(\boldsymbol{t})\geq\varepsilon]\right]$$

$$= \mathbb{E}_{\boldsymbol{t}\sim\boldsymbol{P}}\left[e^{-\theta t_{\max}}|y(\boldsymbol{t})\geq\varepsilon\right]\Pr_{\boldsymbol{t}\sim\boldsymbol{P}}[y(\boldsymbol{t})\geq\varepsilon]$$

$$= \left(\int e^{-\theta x}\mathrm{d}\Pr_{\boldsymbol{t}\sim\boldsymbol{P}}[t_{\max}\leq x]\right)\Pr_{\boldsymbol{t}\sim\boldsymbol{P}}[y(\boldsymbol{t})\geq\varepsilon]$$

$$= -\left(\int \Pr_{\boldsymbol{t}\sim\boldsymbol{P}}[t_{\max}\leq x|y(\boldsymbol{t})\geq\varepsilon]\,\mathrm{d}e^{-\theta x}\right)\Pr_{\boldsymbol{t}\sim\boldsymbol{P}}[y(\boldsymbol{t})\geq\varepsilon] \tag{14}$$

$$= \theta\int \Pr_{\boldsymbol{t}\sim\boldsymbol{P}}[t_{\max}\leq x,y(\boldsymbol{t})\geq\varepsilon]\,e^{-\theta x}\mathrm{d}x \tag{15}$$

where (14) is obtained through integration by parts.

Now, as we can see from (15), the question reduces to bound $\Pr_{\boldsymbol{t}\sim\boldsymbol{P}}[t_{\max}\leq x,y(\boldsymbol{t})\geq\varepsilon]$ for any $x\in\mathbb{R}$. It might be easier to write

$$\Pr_{\boldsymbol{t}\sim\boldsymbol{P}}[t_{\max}\leq x,y(\boldsymbol{t})\geq\varepsilon] = \Pr_{\boldsymbol{t}\sim\boldsymbol{P}}[y(\boldsymbol{t})\geq\varepsilon|t_{\max}\leq x]\Pr_{\boldsymbol{t}\sim\boldsymbol{P}}[t_{\max}\leq x]$$

and we know that

$$\Pr_{\boldsymbol{t}\sim\boldsymbol{P}}[t_{\max}\leq x] = \left(\Pr_{t\sim P}[t\leq x]\right)^k \tag{16}$$

$$= \left((1-q)\Phi(x;0,\sigma^2)+q\Phi(x;1,\sigma^2)\right)^k \tag{17}$$

as all $t_i$s are i.i.d. random samples from $P$, where $\Phi(\cdot;\mu,\sigma^2)$ is the CDF of Gaussian distribution with mean $\mu$ and variance $\sigma^2$.

It remains to bound the conditional probability $\Pr_{\boldsymbol{t}\sim\boldsymbol{P}}[y(\boldsymbol{t})\geq\varepsilon|t_{\max}\leq x]$, it may be easier to see it in this way:

$$\Pr_{\boldsymbol{t}\sim\boldsymbol{P}}[y(\boldsymbol{t})\geq\varepsilon|t_{\max}\leq x]$$

$$= \Pr_{\boldsymbol{t}\sim\boldsymbol{P}}[y(\boldsymbol{t})\geq\varepsilon|t_1\leq x,\ldots,t_k\leq x]$$

$$= \Pr_{\boldsymbol{t}\sim\boldsymbol{P}}\left[\sum_{i=1}^k y(t_i)\geq\varepsilon|y(t_1)\leq y(x),\ldots,y(t_k)\leq y(x)\right]$$

$$\leq \frac{\mathbb{E}_{\boldsymbol{t}\sim\boldsymbol{P}}\left[e^{\lambda\sum_{i=1}^k y(t_i)}\geq e^{\lambda\varepsilon}|y(t_1)\leq y(x),\ldots,y(t_k)\leq y(x)\right]}{e^{\lambda\varepsilon}}$$

where the last step is due to Chernoff bound which holds for any $\lambda>0$. Now we only need to bound the moment generating function for $y(t)=\log(1-q+qe^{\frac{2t-1}{2\sigma^2}}),t\sim P|_{\leq x}$, where $P|_{\leq x}$ is

the *truncated* distribution of $P$. We note that this is equivalent to bounding the Rényi divergence for truncated Gaussian mixture distribution.

Recall that $P = (1-q)\mathcal{N}(0, \sigma^2) + q\mathcal{N}(1, \sigma^2)$. For any $\lambda$ that is a positive integer, we have

$$\mathbb{E}_{t\sim P}\left[e^{\lambda y(t)}I[t \leq x]\right]$$

$$= \frac{1}{\sqrt{2\pi}\sigma}\left[(1-q)\int_{-\infty}^{x}\left(1-q+qe^{\frac{2t-1}{2\sigma^2}}\right)^{\lambda}e^{-\frac{t^2}{2\sigma^2}}dt + q\int_{-\infty}^{x}\left(1-q+qe^{\frac{2t-1}{2\sigma^2}}\right)^{\lambda}e^{-\frac{(t-1)^2}{2\sigma^2}}dt\right]$$

$$= \frac{1}{2}\left(qe^{-\frac{1}{2\sigma^2}}\right)^{\lambda}\sum_{i=0}^{\lambda}\binom{\lambda}{i}\tilde{q}^i\left[(1-q)e^{\frac{(\lambda-i)^i}{2\sigma^2}}\left(\mathrm{erf}\left(\frac{x-(\lambda-i)}{\sqrt{2}\sigma}\right)+1\right)+qe^{\frac{(\lambda-i+1)^i-1}{2\sigma^2}}\left(\mathrm{erf}\left(\frac{x-(\lambda-i+1)}{\sqrt{2}\sigma}\right)+1\right)\right]$$

(18)

where $\tilde{q} := \frac{(1-q)\exp(1/(2\sigma^2))}{q}$. Note that the above expression can be efficiently computed. Denote the above results as $r(\lambda, x) := (18)$. Hence

$$\mathbb{E}_{t\sim P}\left[e^{\lambda y(t)}|t \leq x\right] = \frac{r(\lambda, x)}{\Pr_{t\sim P}[t \leq x]} \tag{19}$$

Now we have

$$\Pr_{t\sim P}[y(t) \geq \varepsilon|t_{\max} \leq x] \leq \frac{\mathbb{E}_{t\sim P}\left[e^{\lambda\sum_{i=1}^{k}y(t_i)} \geq e^{\lambda\varepsilon}|t_{\max} \leq x\right]}{e^{\lambda\varepsilon}}$$

$$= \frac{[r(\lambda, x)]^k}{e^{\lambda\varepsilon}\left(\Pr_{t\sim P}[t \leq x]\right)^k}$$

Plugging this bound into (15), we have

$$(13) = \theta\int\Pr_{t\sim P}[t_{\max} \leq x, y(t) \geq \varepsilon]e^{-\theta x}dx$$

$$\leq \theta e^{-\lambda\varepsilon}\int[r(\lambda, x)]^k e^{-\theta x}dx$$

which leads to the final conclusion. $\square$

**Remark 27.** *In practice, we can further improve the bound by moving the minimum operation inside the integral:*

$$\mathbb{E}[(\widehat{\boldsymbol{\delta}}_{IS,\theta})^2] \leq kM_P(\theta)\theta e^{-\lambda\varepsilon}\int\left[\min_{\lambda}r(\lambda, x)\right]^k e^{-\theta x}dx \tag{20}$$

*Of course, this bound will be less efficient to compute.*

**Theorem 23** (Generalizing Theorem 24 and Theorem 26 via Holder's inequality). *For any positive integer $\lambda$, and for any $a, b \geq 1$ s.t. $1/a + 1/b = 1$, we have $\mathbb{E}[(\widehat{\boldsymbol{\delta}}_{IS,\theta})^2] \leq kM_P(\theta)\left(\mathbb{E}[\widehat{\boldsymbol{\delta}}_{MC}^{2a}]\right)^{1/a} \cdot$ $\left(b\theta e^{-\lambda\varepsilon}\int[r(\lambda, x)]^k e^{-b\theta x}dx\right)^{1/b}$ where $r(\lambda, x)$ is an upper bound for the MGF of privacy loss random variable of* truncated *Gaussian mixture $P|_{\leq x}$ defined in (18).*

*Proof.* Note that Theorem 24 and Theorem 26 can both be viewed as two special cases of Hölder's inequality: for any $a, b \geq 1$ s.t. $\frac{1}{a} + \frac{1}{b} = 1$, we have

$$\mathbb{E}_{t\sim P}\left[\left(1 - e^{\varepsilon-y(t;P,Q)}\right)_{+}^{2}\left(\frac{1}{\sum_{i=1}^{k}e^{\theta t_i}}\right)\right]$$

$$= \mathbb{E}_{t\sim P}\left[\left(1 - e^{\varepsilon-y(t;P,Q)}\right)_{+}^{2}\left(\frac{1}{\sum_{i=1}^{k}e^{\theta t_i}}\right)I[y(t; P, Q) \geq \varepsilon]\right]$$

$$\leq \mathbb{E}_{t\sim P}\left[\left(1 - e^{\varepsilon-y(t;P,Q)}\right)_{+}^{2a}\right]^{1/a}\mathbb{E}_{t\sim P}\left[\left(\frac{1}{\sum_{i=1}^{k}e^{\theta t_i}}\right)^{b}I[y(t; P, Q) \geq \varepsilon]\right]^{1/b}$$

Theorem 24 corresponds to the case where $a = 1$, and Theorem 26 corresponds to the case where $b = 1$. We can actually tune the parameters $a$ and $b$ to see if we can obtain any better bounds, as we have

$$\mathbb{E}_{\boldsymbol{t} \sim \boldsymbol{P}} \left[ \left( 1 - e^{\varepsilon - y(\boldsymbol{t}; \boldsymbol{P}, \boldsymbol{Q})} \right)_+^{2a} \right] \leq \mathbb{E}[\widehat{\boldsymbol{\delta}}_{\mathsf{MC}}^{2a}] \tag{21}$$

and

$$\mathbb{E}_{\boldsymbol{t} \sim \boldsymbol{P}} \left[ \left( \frac{1}{\sum_{i=1}^{k} e^{\theta t_i}} \right)^b I[y(\boldsymbol{t}; \boldsymbol{P}, \boldsymbol{Q}) \geq \varepsilon] \right] \leq b\theta e^{-\lambda \varepsilon} \int [r(\lambda, x)]^k e^{-b\theta x} dx \tag{22}$$

Simply combining the above inequalities leads to the final conclusion. $\qquad \square$

# F Proofs for Utility

## F.1 Overview

While one may be able to bound the false negative rate through similar techniques that we bound the false positive rate, i.e., applying the concentration inequalities, the guarantee may be loose. As a formal, strict guarantee for $\mathsf{FN}_{\mathsf{DPV}}$ is not required, we provide a convenient heuristic of picking an appropriate $\Delta$ such that $\mathsf{FN}_{\mathsf{DPV}}$ is *approximately* smaller than $\mathsf{FP}_{\mathsf{DPV}}$.

For any mechanism $\mathcal{M}$ such that $\delta^{\mathrm{est}} > \rho \delta_Y(\varepsilon)$, we have

$$
\Pr_{\mathsf{DPV}} \left[ \mathsf{DPV}(\mathcal{M}, \varepsilon, \delta^{\mathrm{est}}) = \texttt{False} \right]
$$
$$
= \Pr \left[ \widehat{\boldsymbol{\delta}}_{\mathsf{MC}}^m > \delta^{\mathrm{est}}/\tau - \Delta \right]
$$
$$
= \Pr \left[ \widehat{\boldsymbol{\delta}}_{\mathsf{MC}}^m - \delta_Y(\varepsilon) > \delta^{\mathrm{est}}/\tau - \delta_Y(\varepsilon) - \Delta \right]
$$
$$
\leq \Pr \left[ \widehat{\boldsymbol{\delta}}_{\mathsf{MC}}^m - \delta_Y(\varepsilon) > \left( (1/\tau - 1/\rho) \, \delta^{\mathrm{est}} - \Delta \right) \right]
$$

and in the meantime, if $\delta^{\mathrm{est}} < \tau \delta_Y(\varepsilon)$, we have

$$
\Pr \left[ \widehat{\boldsymbol{\delta}}_{\mathsf{MC}}^m < \delta^{\mathrm{est}}/\tau - \Delta \right] \leq \Pr \left[ \widehat{\boldsymbol{\delta}}_{\mathsf{MC}}^m - \delta_Y(\varepsilon) < -\Delta \right]
$$

Our main idea is to find $\Delta$ such that

$$
\Pr \left[ \widehat{\boldsymbol{\delta}}_{\mathsf{MC}}^m - \delta_Y(\varepsilon) > \left( (1/\tau - 1/\rho) \, \delta^{\mathrm{est}} - \Delta \right) \right]
$$
$$
\lesssim \Pr \left[ \widehat{\boldsymbol{\delta}}_{\mathsf{MC}}^m - \delta_Y(\varepsilon) < -\Delta \right]
$$

In this way, we know that $\mathsf{FN}_{\mathsf{MC}}(\varepsilon, \delta^{\mathrm{est}}; \rho)$ is upper bounded by $\Theta(\delta^{\mathrm{est}}/\tau)$ for the same amount of samples we discussed in Section 4.3.

Observe that $\widehat{\boldsymbol{\delta}}_{\mathsf{MC}}$ (or $\widehat{\boldsymbol{\delta}}_{\mathsf{IS},\theta}$ with a not too large $\theta$) is typically a highly asymmetric distribution with a significant probability of being zero, and the probability density decreases monotonically for higher values. Under such conditions, we prove the following results:

**Theorem 28** (Informal). *When $m \geq \frac{2\nu}{\Delta^2} \log(\tau/\delta^{\mathrm{est}})$, we have*

$$
\Pr[\widehat{\boldsymbol{\delta}}_{\mathsf{MC}}^m - \delta_Y(\varepsilon) < -\Delta] \gtrsim \Pr[\widehat{\boldsymbol{\delta}}_{\mathsf{MC}}^m - \delta_Y(\varepsilon) > \frac{3}{2}\Delta]
$$

The proof is deferred to Appendix F.2. Therefore, by setting $\Delta = 0.4 \, (1/\tau - 1/\rho) \, \delta^{\mathrm{est}}$, one can ensure that $\mathsf{FN}_{\mathsf{MC}}(\varepsilon, \delta^{\mathrm{est}}; \rho)$ is also (approximately) upper bounded by $\Theta(\delta^{\mathrm{est}}/\tau)$. We empirically verify the effectiveness of such a heuristic by estimating the actual false negative rate. As we can see from Figure 6 (b), the dashed curve is much higher than the two solid curves, which means that the false negative rate is a very conservative bound.

**Remark 29** (For the halting case). *In this section, we develop techniques for ensuring the false negative rate (i.e., the rejection probability) is around $O(\delta)$ when the proposed privacy parameter $\delta^{\mathrm{est}}$ is close to the true $\delta$. In the experiment, we use the state-of-the-art FFT accountant to produce $\delta^{\mathrm{est}}$, which is very accurate as we can see from Figure 1. Hence, the rejection probability in the experiment is around $O(\delta)$, which means the probability of rejection is close to the probability of catastrophic failure for privacy.*

*If one is still concerned that the rejection probability is too large, we can further reduce the probability as follows: we run two instances of EVR paradigm simultaneously; if both of the instances are passed, we randomly pick one and release the output. If either one of them is passed, we release the passed instance. It only fails when both of the instances fail. By running two instances of the EVR paradigm in parallel, the false positive rate (i.e., the final $\delta$) will be only doubled, but the probability of rejection will be squared.*

*We can also introduce a variant of our EVR paradigm that better deals with the failure case: whenever we face the "rejection", we run a different mechanism $\mathcal{M}'$ that is guaranteed to be $(\varepsilon, \delta^{\mathrm{est}})$-DP*

*(e.g., by adjusting the subsampling rate and/or noise multiplier in DP-SGD). Moreover, we use FFT accountant to obtain a strict privacy guarantee upper bound $(\varepsilon, \delta^{\mathrm{upp}})$ for the original mechanism $\mathcal{M}$, where $\delta^{\mathrm{est}} < \delta^{\mathrm{upp}}$. We use $p_{\mathsf{FN}}$ and $p_{\mathsf{FP}}$ to denote the false negative and false positive rate of the privacy verifier used in EVR paradigm. If the original mechanism $\mathcal{M}$ is indeed $(\varepsilon, \delta^{\mathrm{est}})$-DP, then for any subset $S$, for this augmented EVR paradigm $\mathcal{M}^{\mathrm{aug}}$ we have*

$$
\begin{aligned}
\Pr[\mathcal{M}^{\mathrm{aug}}(D) \in S] &= p_{\mathsf{FN}} \Pr[\mathcal{M}(D) \in S] + (1 - p_{\mathsf{FN}}) \Pr[\mathcal{M}'(D) \in S] \\
&\leq p_{\mathsf{FN}}(e^\varepsilon \Pr[\mathcal{M}(D') \in S] + \delta^{\mathrm{est}}) + (1 - p_{\mathsf{FN}})(e^\varepsilon \Pr[\mathcal{M}'(D') \in S] + \delta^{\mathrm{est}}) \\
&\leq e^\varepsilon \Pr[\mathcal{M}^{\mathrm{aug}}(D') \in S] + \delta^{\mathrm{est}}
\end{aligned}
$$

*If the original mechanism $\mathcal{M}$ is not $(\varepsilon, \delta^{\mathrm{est}})$-DP, then we have*

$$
\begin{aligned}
\Pr[\mathcal{M}^{\mathrm{aug}}(D) \in S] &= p_{\mathsf{FP}} \Pr[\mathcal{M}(D) \in S] + (1 - p_{\mathsf{FP}}) \Pr[\mathcal{M}'(D) \in S] \\
&\leq p_{\mathsf{FP}}(e^\varepsilon \Pr[\mathcal{M}(D') \in S] + \delta^{\mathrm{upp}}) + (1 - p_{\mathsf{FP}})(e^\varepsilon \Pr[\mathcal{M}'(D') \in S] + \delta^{\mathrm{est}}) \\
&\leq e^\varepsilon \Pr[\mathcal{M}^{\mathrm{aug}}(D') \in S] + \delta^{\mathrm{est}} + p_{\mathsf{FP}}(\delta^{\mathrm{upp}} - \delta^{\mathrm{est}})
\end{aligned}
$$

*Hence, this augmented EVR algorithm will be $(\varepsilon, \delta^{\mathrm{est}} + p_{\mathsf{FP}}(\delta^{\mathrm{upp}} - \delta^{\mathrm{est}}))$, and if $p_{\mathsf{FP}}$ is around $\delta^{\mathrm{est}}$, then this extra factor $p_{\mathsf{FP}}(\delta^{\mathrm{upp}} - \delta^{\mathrm{est}})$ will be very small. We can also adjust the privacy guarantee for $\mathcal{M}'$ such that the privacy guarantees for the two cases are the same, which can further optimize the final privacy cost.*

## F.2 Technical Details

In this section, we provide theoretical justification for the heuristic of setting $\Delta = 0.4 \left( 1/\tau - 1/\rho \right) \delta^{\text{est}}$.

For notation convenience, throughout this section, we talk about $\widehat{\delta}_{\text{MC}}$, but the same argument also applies to $\widehat{\delta}_{\text{IS},\theta}$ with a not too large $\theta$, unless otherwise specified. We use $\widehat{\delta}_{\text{MC}}(x)$ to denote the density of $\widehat{\delta}_{\text{MC}}$ at $x$. Note that $\widehat{\delta}_{\text{MC}}(0) = \infty$.

We make the following assumption about the distribution of $\widehat{\delta}_{\text{MC}}$.

**Assumption 30.** $\Pr[\widehat{\delta}_{MC} = 0] \geq 1/2$.

While intuitive, this assumption is hard to analyze for the case of Subsampled Gaussian mechanism. Therefore, we instead provide a condition for which the assumption holds for Pure Gaussian mechanism.

**Lemma 31.** *Fix $\varepsilon, \sigma$. When $k/(2\sigma^2) \leq \varepsilon$, Assumption 30 holds.*

*Proof.* The PRV for the composition of $k$ Gaussian mechanism is $\mathcal{N}\left( \frac{k}{2\sigma^2}, \frac{k}{\sigma^2} \right)$.

$$\Pr[(1 - e^{\varepsilon - Y})_+ = 0] = \Pr[Y \leq \varepsilon]$$

which is clearly $\geq 1/2$ when $k/(2\sigma^2) \leq \varepsilon$. $\qquad\square$

We also empirically verify this assumption for Subsampled Gaussian mechanism as in Figure 7. As we can see, the $\theta$ selected by our heuristic (the red star) has $\Pr[\widehat{\delta}_{\text{IS},\theta} = 0] \approx 1/2$ which matches our principle of selecting $\theta$. The $\theta$ minimizes the analytical bound (**which we are going to use in practice**) achieves $\Pr[\widehat{\delta}_{\text{IS},\theta} = 0] \approx 0.88 \gg 0.5$.

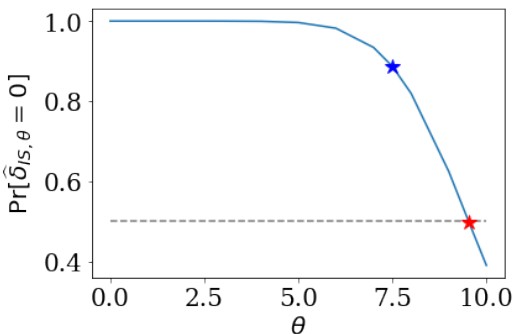

Figure 7: We empirically estimate $\Pr[\widehat{\delta}_{\text{IS},\theta} = 0]$ for the case of Poisson Subampled Gaussian mechanism. We set $q = 10^{-3}, \sigma = 0.6, \varepsilon = 1.5, k = 100$. $\delta_Y(\varepsilon) \approx 7.7 \times 10^{-6}$ in this case. The red star indicates the second moment for the value of $\theta$ selected by our heuristic in Definition 18. The blue star indicates the $\theta$ that minimizes the analytical bound.

Our goal is to show that $\Pr[\widehat{\delta}_{\text{MC}}^m - \delta_Y(\varepsilon) < -\Delta^*] \gtrsim \Pr[\widehat{\delta}_{\text{MC}}^m - \delta_Y(\varepsilon) > \Delta^*]$ for large $m$. For notation simplicity, we denote $\delta := \mathbb{E}[\widehat{\delta}_{\text{MC}}] = \delta_Y(\varepsilon)$ in the remaining of the section. We also denote

$$p_0 := \Pr[\widehat{\delta}_{\text{MC}} = 0]$$
$$p_{(0,1)} := \Pr[0 < \widehat{\delta}_{\text{MC}} < \delta]$$
$$p_{(1,2)} := \Pr[\delta \leq \widehat{\delta}_{\text{MC}} \leq 2\delta]$$
$$p_{(2,\infty)} := \Pr[\widehat{\delta}_{\text{MC}} \geq 2\delta]$$

Note that $p_0 + p_{(0,1)} + p_{(1,2)} + p_{(2,\infty)} = 1$. We also write $\widehat{\delta}_{\text{MC}}|_{(a,b)}$ and $\widehat{\delta}_{\text{MC}}|_{[a,b)}$ to indicate the truncated distribution of $\widehat{\delta}_{\text{MC}}$ on $(a,b)$ and $[a,b)$, respectively.

We first construct an alternative random variable $\widetilde{\boldsymbol{\delta}}_{\text{MC}}$ with the following distribution:

$$
\widetilde{\boldsymbol{\delta}}_{\text{MC}} = \begin{cases}
2\delta - x & \text{for } x \sim \widehat{\boldsymbol{\delta}}_{\text{MC}}|_{\geq 2\delta} & \text{w.p. } p_{(2,\infty)} \\
x & \text{for } x \sim \widehat{\boldsymbol{\delta}}_{\text{MC}}|_{(0,\delta)} & \text{w.p. } p_{(0,1)} \\
2\delta - x & \text{for } x \sim \widehat{\boldsymbol{\delta}}_{\text{MC}}|_{(0,\delta)} & \text{w.p. } p_{(0,1)} \\
x & \text{for } x \sim \widehat{\boldsymbol{\delta}}_{\text{MC}}|_{\geq 2\delta} & \text{w.p. } p_{(2,\infty)} \\
\delta & & \text{w.p. } 1 - 2(p_{(0,1)} + p_{(2,\infty)})
\end{cases}
\tag{23}
$$

This is a valid probability due to Assumption 30, as $p_{(0,1)} + p_{(2,\infty)} \leq 1 - p_0 \leq 1/2$. The distribution of $\widetilde{\boldsymbol{\delta}}_{\text{MC}}$ is illustrated in Figure 8. Note that $\widetilde{\boldsymbol{\delta}}_{\text{MC}}$ is a symmetric distribution with $\mathbb{E}[\widetilde{\boldsymbol{\delta}}_{\text{MC}}] = \delta$.

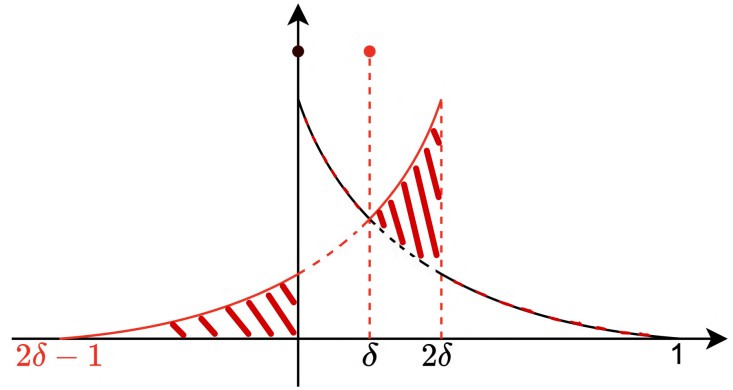

Figure 8: Illustration for the density function of $\widetilde{\boldsymbol{\delta}}_{\text{MC}}$ (black curve indicates the density curve for $\widehat{\boldsymbol{\delta}}_{\text{MC}}$, and red curve indicates the density curve for $\widehat{\boldsymbol{\delta}}_{\text{IS},\theta}$).

Similar to the notation of $\widehat{\boldsymbol{\delta}}_{\text{MC}}^m$, we write $\widetilde{\boldsymbol{\delta}}_{\text{MC}}^m := \frac{1}{m} \sum_{i=1}^{m} \widetilde{\boldsymbol{\delta}}_{\text{MC}}^{(i)}$. We show an asymmetry result for $\widetilde{\boldsymbol{\delta}}_{\text{MC}}^m$ in terms of $\delta$.

**Lemma 32.** *For any $\Delta^* \in \mathbb{R}$, we have*

$$\Pr[\widetilde{\boldsymbol{\delta}}_{MC}^m - \delta < -\Delta^*] \geq \Pr[\widetilde{\boldsymbol{\delta}}_{MC}^m - \delta > \Delta^*] \tag{24}$$

*Proof.* Note that the above argument holds trivially when $m = 0, 1$. For $m \geq 2$, we use the induction argument. Suppose we have

$$\Pr[\widetilde{\boldsymbol{\delta}}_{\text{MC}}^{m-1} - \delta < -\Delta^*] \geq \Pr[\widetilde{\boldsymbol{\delta}}_{\text{MC}}^{m-1} - \delta > \Delta^*] \tag{25}$$

for any $\Delta^* \in \mathbb{R}$. That is,

$$\Pr\left[\sum_{i=1}^{m-1} \widetilde{\boldsymbol{\delta}}_{\text{MC}}^{(i)} - (m-1)\delta < -(m-1)\Delta^*\right] \geq \Pr\left[\sum_{i=1}^{m-1} \widetilde{\boldsymbol{\delta}}_{\text{MC}}^{(i)} - (m-1)\delta > (m-1)\Delta^*\right] \tag{26}$$

for any $\Delta^* \in \mathbb{R}$.

$$\begin{aligned}
\Pr[\widetilde{\boldsymbol{\delta}}_{\text{MC}}^m - \delta < -\Delta^*] &= \Pr\left[\sum_{i=1}^{m} \widetilde{\boldsymbol{\delta}}_{\text{MC}}^{(i)} - m\delta < -m\Delta^*\right] \\
&= \Pr\left[\sum_{i=1}^{m-1} \widetilde{\boldsymbol{\delta}}_{\text{MC}}^{(i)} - (m-1)\delta + \widetilde{\boldsymbol{\delta}}_{\text{MC}}^{(m)} - \delta < -m\Delta^*\right] \\
&= \int \Pr\left[\sum_{i=1}^{m-1} \widetilde{\boldsymbol{\delta}}_{\text{MC}}^{(i)} - (m-1)\delta < -m\Delta^* - x\right] \Pr[\widetilde{\boldsymbol{\delta}}_{\text{MC}}^{(m)} - \delta = x]\mathrm{d}x \\
&\geq \int \Pr\left[\sum_{i=1}^{m-1} \widetilde{\boldsymbol{\delta}}_{\text{MC}}^{(i)} - (m-1)\delta > m\Delta^* + x\right] \Pr[\widetilde{\boldsymbol{\delta}}_{\text{MC}}^{(m)} - \delta = x]\mathrm{d}x \\
&= \Pr\left[\sum_{i=1}^{m-1} \widetilde{\boldsymbol{\delta}}_{\text{MC}}^{(i)} - (m-1)\delta - (\widetilde{\boldsymbol{\delta}}_{\text{MC}}^{(m)} - \delta) > m\Delta^*\right] \\
&= \int \Pr\left[x - (\widetilde{\boldsymbol{\delta}}_{\text{MC}}^{(m)} - \delta) > m\Delta^*\right] \Pr\left[\sum_{i=1}^{m-1} \widetilde{\boldsymbol{\delta}}_{\text{MC}}^{(i)} - (m-1)\delta = x\right]\mathrm{d}x \\
&= \int \Pr\left[\widetilde{\boldsymbol{\delta}}_{\text{MC}}^{(m)} - \delta < -m\Delta^* + x\right] \Pr\left[\sum_{i=1}^{m-1} \widetilde{\boldsymbol{\delta}}_{\text{MC}}^{(i)} - (m-1)\delta = x\right]\mathrm{d}x \\
&\geq \int \Pr\left[\widetilde{\boldsymbol{\delta}}_{\text{MC}}^{(m)} - \delta > m\Delta^* - x\right] \Pr\left[\sum_{i=1}^{m-1} \widetilde{\boldsymbol{\delta}}_{\text{MC}}^{(i)} - (m-1)\delta = x\right]\mathrm{d}x \\
&= \left[\sum_{i=1}^{m} \widetilde{\boldsymbol{\delta}}_{\text{MC}}^{(i)} - m\delta > m\Delta^*\right] \\
&= \left[\widetilde{\boldsymbol{\delta}}_{\text{MC}}^m - \delta > \Delta^*\right]
\end{aligned}$$

where the two inequalities are due to the induction assumption. $\qquad\square$

Now we come back and analyze $\widehat{\boldsymbol{\delta}}_{\text{MC}}^m$ by using Lemma 32.

$$\Pr\left[\widehat{\boldsymbol{\delta}}_{\text{MC}}^m - \delta \leq -\Delta^*\right]$$

$$= \Pr\left[\widehat{\boldsymbol{\delta}}_{\text{MC}}^m - \widetilde{\boldsymbol{\delta}}_{\text{MC}}^m + \widetilde{\boldsymbol{\delta}}_{\text{MC}}^m - \delta \leq -\Delta^*\right]$$

$$\geq \Pr\left[\widehat{\boldsymbol{\delta}}_{\text{MC}}^m - \widetilde{\boldsymbol{\delta}}_{\text{MC}}^m \leq c\right] \Pr\left[\widehat{\boldsymbol{\delta}}_{\text{MC}}^m - \widetilde{\boldsymbol{\delta}}_{\text{MC}}^m + \widetilde{\boldsymbol{\delta}}_{\text{MC}}^m - \delta \leq -\Delta^* | \widehat{\boldsymbol{\delta}}_{\text{MC}}^m - \widetilde{\boldsymbol{\delta}}_{\text{MC}}^m \leq c\right]$$

$$\geq \Pr\left[\widehat{\boldsymbol{\delta}}_{\text{MC}}^m - \widetilde{\boldsymbol{\delta}}_{\text{MC}}^m \leq c\right] \Pr\left[\widetilde{\boldsymbol{\delta}}_{\text{MC}}^m - \delta \leq -\Delta^* - c\right]$$

$$\geq \Pr\left[\widehat{\boldsymbol{\delta}}_{\text{MC}}^m - \widetilde{\boldsymbol{\delta}}_{\text{MC}}^m \leq c\right] \Pr\left[\widetilde{\boldsymbol{\delta}}_{\text{MC}}^m - \delta \geq \Delta^* + c\right]$$

Similarly,

$$\Pr\left[\widetilde{\boldsymbol{\delta}}_{\text{MC}}^m - \delta \geq \Delta^* + c\right]$$

$$= \Pr\left[\widetilde{\boldsymbol{\delta}}_{\text{MC}}^m - \widehat{\boldsymbol{\delta}}_{\text{MC}}^m + \widehat{\boldsymbol{\delta}}_{\text{MC}}^m - \delta \geq \Delta^* + c\right]$$

$$\geq \Pr\left[\widetilde{\boldsymbol{\delta}}_{\text{MC}}^m - \widehat{\boldsymbol{\delta}}_{\text{MC}}^m \geq -c\right] \Pr\left[\widetilde{\boldsymbol{\delta}}_{\text{MC}}^m - \widehat{\boldsymbol{\delta}}_{\text{MC}}^m + \widehat{\boldsymbol{\delta}}_{\text{MC}}^m - \delta \geq \Delta^* + c | \widetilde{\boldsymbol{\delta}}_{\text{MC}}^m - \widehat{\boldsymbol{\delta}}_{\text{MC}}^m \geq -c\right]$$

$$\geq \Pr\left[\widetilde{\boldsymbol{\delta}}_{\text{MC}}^m - \widehat{\boldsymbol{\delta}}_{\text{MC}}^m \geq -c\right] \Pr\left[\widehat{\boldsymbol{\delta}}_{\text{MC}}^m - \delta \geq \Delta^* + 2c\right]$$

Overall, we have

$$\Pr\left[\widehat{\boldsymbol{\delta}}_{\text{MC}}^m - \delta \leq -\Delta^*\right] \geq \left(\Pr\left[\widehat{\boldsymbol{\delta}}_{\text{MC}}^m - \widetilde{\boldsymbol{\delta}}_{\text{MC}}^m \leq c\right]\right)^2 \Pr\left[\widehat{\boldsymbol{\delta}}_{\text{MC}}^m - \delta \geq \Delta^* + 2c\right] \tag{27}$$

for any $\Delta^* \in \mathbb{R}$. Note that the above argument does not require $\widehat{\boldsymbol{\delta}}_{\text{MC}}^m$ and $\widetilde{\boldsymbol{\delta}}_{\text{MC}}^m$ to be correlated. To maximize $\Pr\left[\widehat{\boldsymbol{\delta}}_{\text{MC}}^m - \widetilde{\boldsymbol{\delta}}_{\text{MC}}^m \leq c\right]$, we can sample $\widetilde{\boldsymbol{\delta}}_{\text{MC}}^m$ for a given $\widehat{\boldsymbol{\delta}}_{\text{MC}}^m$ as follows: for each $\widehat{\boldsymbol{\delta}}_{\text{MC}}^{(i)}$,

1. If $\widehat{\boldsymbol{\delta}}_{\text{MC}}^{(i)} > 0$, then let $\widetilde{\boldsymbol{\delta}}_{\text{MC}}^{(i)} = \widehat{\boldsymbol{\delta}}_{\text{MC}}^{(i)}$.

2. If $\widehat{\boldsymbol{\delta}}_{\text{MC}}^{(i)} = 0$, then with probability $p_{(2,\infty)}/p_0$, output $2\delta - x$ for $x \sim \widehat{\boldsymbol{\delta}}_{\text{MC}}|_{(2,\infty)}$; with probability $p_{(0,1)}/p_0$ output $2\delta - x$ for $x \sim \widehat{\boldsymbol{\delta}}_{\text{MC}}|_{(0,1)}$; with probability $1 - (p_{(0,1)} + p_{(2,\infty)})/p_0$ output $\delta$.

Denote the random variable $\boldsymbol{\delta}_{\text{diff}} := \widehat{\boldsymbol{\delta}}_{\text{MC}} - \widetilde{\boldsymbol{\delta}}_{\text{MC}}$. It is not hard to see that $\mathbb{E}[\boldsymbol{\delta}_{\text{diff}}^2] \leq \mathbb{E}[\widehat{\boldsymbol{\delta}}_{\text{MC}}^2] + \delta^2 \leq 2\mathbb{E}[\widehat{\boldsymbol{\delta}}_{\text{MC}}^2]$. By Bennett's inequality, with $m \geq \frac{2\nu}{\Delta^2} \log(\tau/\delta^{\text{est}})$, we have

$$\Pr\left[\widehat{\boldsymbol{\delta}}_{\text{MC}}^m - \widetilde{\boldsymbol{\delta}}_{\text{MC}}^m \leq c\right] \gtrsim 1 - \frac{\tau}{\delta^{\text{est}}} \exp\left(-\frac{c^2}{\Delta^2}\right)$$

$$= 1 - O\left(\frac{\tau}{\delta^{\text{est}}}\right)$$

if we set $c = O(\Delta)$. Let $c = \frac{1}{4}\Delta$, then by setting $-\Delta^* = -\Delta$ and $\Delta^* + 2c = (1/\tau - 1/\rho)\,\delta^{\text{est}} - \Delta$, we have $\Delta = 0.4\,(1/\tau - 1/\rho)\,\delta^{\text{est}}$.

To summarize, when we set $\Delta$ with the heuristic, we have

$$\Pr\left[\widehat{\boldsymbol{\delta}}_{\text{MC}}^m - \delta \geq \Delta^* + 2c\right] \lesssim \frac{\tau/\delta^{\text{est}}}{\left(1 - O\left(\frac{\tau}{\delta^{\text{est}}}\right)\right)^2}$$

$$\approx \tau/\delta^{\text{est}}$$

# G   Pseudocode for Privacy Accounting for DP-SGD with EVR Paradigm

In this section, we outline the steps of privacy accounting for DP-SGD with our EVR paradigm. Recall from Lemma 16 that for subsampled Gaussian mechanism with sensitivity $C$, noise variance $C^2\sigma^2$, and subsampling rate $q$, one dominating pair $(P, Q)$ is $Q := \mathcal{N}(0, \sigma^2)$ and $P := (1 - q)\mathcal{N}(0, \sigma^2) + q\mathcal{N}(1, \sigma^2)$. Hence, for DP-SGD with $k$ iterations, the dominating pair is the product distribution $\boldsymbol{P} := P_1 \times \ldots \times P_k$ and $\boldsymbol{Q} := Q_1 \times \ldots \times Q_k$ where each $P_i$ and $Q_i$ follow the same distribution as $P$ and $Q$.[3]

---

**Algorithm 4** Privacy Accounting for DP-SGD with EVR Paradigm

---

1: **Privacy Parameters for DP-SGD:** $k$ – number of mechanisms to be composed, $\sigma$ – noise multiplier, $q$ – subsampling rate, $C$ – clipping ratio.

2:

3: `// Step 1:  Estimate Privacy Parameter.`

4: Obtain a privacy parameter estimate $(\varepsilon, \delta^{\text{est}})$ from a DP accountant (e.g., FFT/GDP/MC accountant). Set the smoothing factor $\tau = 0.99$ (one can also adjust the value of $\tau$ according to the privacy guarantee and runtime requirements).

5:

6: `// Step 2:  Verify Privacy Parameter.`

7:

8: `// Step 2.1:  Derive the required amount of samples for privacy verification.`

9: Compute the number of required samples $m$ according to Theorem 13, where the second moment upper bound $\nu$ is computed according to Corollary 22 for Simple MC estimator, or Theorem 23 for Importance Sampling estimator. Set $\rho = (1 + \tau)/2$ and set $\Delta$ according to Theorem 14 for utility guarantee.

10:

11: `// Step 2.2:  Estimate privacy parameter with MC estimator.`

12: If use Simple MC estimator: Obtain i.i.d. samples $\{\boldsymbol{t}_i\}_{i=1}^m$ from $\boldsymbol{P}$. Compute simple MC estimate $\widehat{\delta} = \frac{1}{m} \sum_{i=1}^m \left(1 - e^{\varepsilon - y_i}\right)_+$ with PRV samples $y_i = \log\left(\frac{\boldsymbol{P}(\boldsymbol{t}_i)}{\boldsymbol{Q}(\boldsymbol{t}_i)}\right), i = 1 \ldots m$.

13: If use Importance Sampling estimator: compute MC estimate according to Definition 18.

14:

15: `// Step 3:  Release according to verification result.`

16: **if** $\widehat{\delta} < \frac{\delta^{\text{est}}}{\tau} - \Delta$ **then** release the result of DP-SGD.

17: **else** terminate program.

---

---

[3]It can also be extended to the heterogeneous case easily.

# H Experiment Settings & Additional Results

## H.1 GPU Acceleration for MC Sampling

MC-based techniques are well-suited for parallel computing and GPU acceleration due to their nature of repeated sampling. One can easily utilize PyTorch's CUDA functionality, e.g.,

```
torch.randn(size=(k,m)).cuda()*sigma+mu
```

, to significantly boost the computational efficiency. Figure 9 (a) shows that when using a NVIDIA A100-SXM4-80GB GPU, the execution time of sampling Gaussian mixture $((1 - q)\mathcal{N}(0, \sigma^2) + q\mathcal{N}(1, \sigma^2))$ can be improved by $10^3$ times compared with CPU-only scenario. Figure 9 (b) shows the predicted runtime for different target false positive rates for $k = 1000$. We vary $\sigma$ and set the target false positive rate as the smallest $s \times 10^{-r}$ that is greater than $\delta_Y(\varepsilon)$, where $s \in \{1, 5\}$ and $r$ is positive integer. We set $\delta^{\text{est}} = 0.8\delta_Y(\varepsilon)$, and $\tau, \rho, \Delta$ are set as the heuristics introduced in the previous sections. The runtime is predicted by the number of required samples for the given false positive rate. As we can see, even when we target at $10^{-10}$ false positive rate (which means that $\Delta \approx 10^{-10}$), the clock time is still acceptable (around 3 hours).

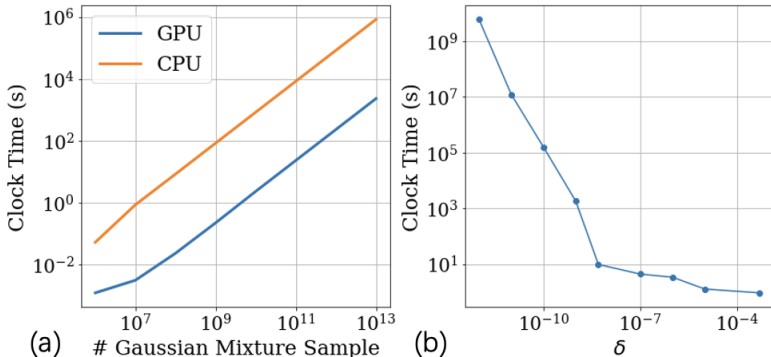

(a)  (b)

Figure 9: The execution time of sampling Gaussian mixture $((1 - q)\mathcal{N}(0, \sigma^2) + q\mathcal{N}(1, \sigma^2))$ when only using CPU and when using a NVIDIA A100-SXM4-80GB GPU.

### H.2 Experiment for Evaluating EVR Paradigm

#### H.2.1 Settings

For Figure 1 & Figure 3, the FFT-based method has hyperparameter being set as $\varepsilon_{\text{error}} = 10^{-3}, \delta_{\text{error}} = 10^{-10}$. For the GDP-Edgeworth accountant, we use the second-order expansion and uniform bound, following [41].

For Figure 4 (as well as Figure 11), the BEiT [4] is first self-supervised pretrained on ImageNet-1k and then trained finetuned on ImageNet-21k, following the state-of-the-art approach in [34]. For DP-GD training, we set $\sigma$ as 28.914, clipping norm as 1, learning rate as 2, and we train for at most 60 iterations, and we only finetune the last layer on CIFAR-100.

#### H.2.2 Additional Results

$k \to \varepsilon_{Y^{(k)}}(\delta)$ **curve.** We show additional results for a more common setting in privacy-preserving machine learning where one set a target $\delta$ and try to find $\varepsilon_{Y^{(k)}}(\delta)$ for different $k$, the number of individual mechanisms in the composition. We use $Y^{(k)}$ to stress that the PRV $Y$ is for the composition of $k$ mechanisms. Such a scenario can happen when one wants to find the optimal stopping iteration for training a differentially private neural network.

Figure 10 shows such a result for Poisson Subsampled Gaussian where we set subsampling rate 0.01, $\sigma = 2$, and $\delta = 10^{-5}$. We set $\varepsilon_{\text{error}} = 10^{-1}, \delta_{\text{error}} = 10^{-10}$ for the FFT method. The estimate in this case is obtained by fixing $\delta^{\text{est}} = \tau\delta$ and find the corresponding estimate for $\varepsilon$ through FFT-based method [19]. As we can see, the EVR paradigm achieves a much tighter privacy analysis compared with the upper bound derived by FFT-based method. The runtime of privacy verification in this case is $< 15$ minutes for all $k$s.

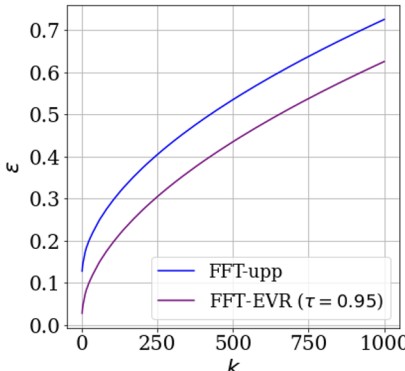

Figure 10: $k \to \varepsilon_{Y^{(k)}}(\delta)$ curve for Poisson Subsampled Gaussian mechanism for subsampling rate 0.01, $\sigma = 2$, and $\delta = 10^{-5}$. When running on an NVIDIA A100-SXM4-80GB GPU, the runtime of privacy verification is $< 15$ minutes.

**Privacy-Utility Tradeoff.** We show additional results for the privacy-utility tradeoff curve when finetuning ImageNet-pretrained BEiT on CIFAR100 dataset with DP stochastic gradient descent (DP-SGD). For DP-SGD training, we set $\sigma$ as 5.971, clipping norm as 1, learning rate as 0.2, momentum as 0.9, batch size as 4096, and we train for at most 360 iterations (30 epochs). We only finetune the last layer on CIFAR-100.

As shown in Figure 11 (a), the EVR paradigm provides a better utility-privacy tradeoff compare with the traditional upper bound method. In Figure 11 (b), we show the runtime of DP verification when $\rho = (1 + \tau)/2$ and we set $\Delta$ according to Theorem 14 (which ensures EVR's failure probability is negligible). The runtime is estimated on an NVIDIA A100-SXM4-80GB GPU. As we can see, it only takes a few minutes for privacy verification, which is short compared with hours of model training.

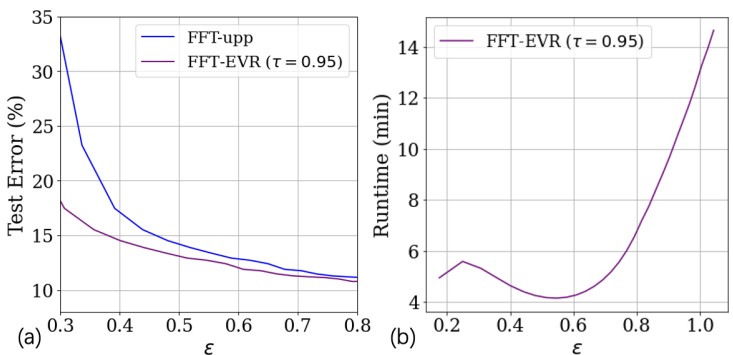

Figure 11: (a) Utility-privacy tradeoff curve for fine-tuning ImageNet-pretrained BEiT [4] on CIFAR100 when $\delta = 10^{-12}$ with DP-SGD. We follow the training hyperparameters from [34]. (b) Runtime of privacy verification in the EVR paradigm. For fair comparison, we set $\rho = (1 + \tau)/2$ and set $\Delta$ according to Theorem 14, which ensures EVR's failure probability is around $O(\delta)$. For (b), the runtime is estimated on an NVIDIA A100-SXM4-80GB GPU.

## H.3 Experiment for Evaluating MC Accountant

### H.3.1 Settings

**Evaluation Protocol.** We use $Y^{(k)}$ to stress that the PRV $Y$ is for the composition of $k$ Poisson Subsampled Gaussian mechanisms. For the offline setting, we make the following two kinds of plots: **(1)** the relative error in approximating $\varepsilon \mapsto \delta_{Y^{(k)}}(\varepsilon)$ (for fixed $k$), and **(2)** the relative error in $k \mapsto \varepsilon_{Y^{(k)}}(\delta)$ (for fixed $\delta$), where $\varepsilon_{Y^{(k)}}(\delta)$ is the inverse of $\delta_{Y^{(k)}}(\varepsilon)$ from (2). For the online setting, we make the following two kinds of plots: **(1)** the relative error in approximating $k \mapsto \varepsilon_{Y^{(k)}}(\delta)$ (for fixed $\delta$), and **(2)** $k \mapsto$ cumulative time for privacy accounting until $k$th iteration.

**MC Accountant.** We use the importance sampling technique with the tilting parameter being set according to the heuristic described in Definition 18.

**Baselines.** We compare MC accountant against the following state-of-the-art DP accountants with the following settings:

- The state-of-the-art FFT-based approach [19]. The setting of $\varepsilon_{\text{error}}$ and $\delta_{\text{error}}$ is specified in the next section.
- CLT-based GDP accountant [7].
- GDP-Edgeworth accountant with second-order expansion and uniform bound.
- The Analytical Fourier Accountant based on characteristic function [44], with double quadrature approximation as this is the practical method recommended in the original paper.

### H.3.2 Additional Results for Online Accounting

Figure 12 and 13 show the online accounting results for $(\sigma, \delta, q) = (0.5, 10^{-5}, 10^{-3})$ and $(\sigma, \delta, q) = (0.5, 10^{-13}, 10^{-3})$, respectively. For the setting of $(\sigma, \delta, q) = (0.5, 10^{-5}, 10^{-3})$, we can see that the MC accountant achieves a comparable performance with a shorter runtime. For the setting of $(\sigma, \delta, q) = (0.5, 10^{-13}, 10^{-3})$, we can see that the MC accountant achieves significantly better performance compared to the state-of-the-art FFT accountant (and again, with a shorter runtime). This showcases the MC accountant's efficiency and accuracy in online setting.

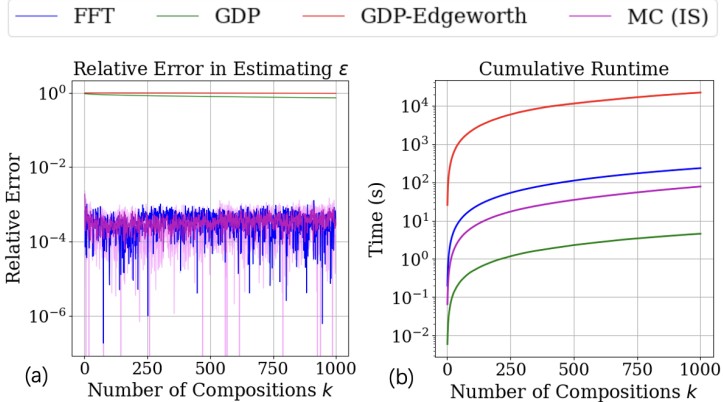

Figure 12: Experiment for Composing Subsampled Gaussian Mechanisms in the Online Setting with hyperparameter $(\sigma, \delta, q) = (0.5, 10^{-5}, 10^{-3})$. (a) Compares the relative error in approximating $k \mapsto \varepsilon_Y(\delta)$. The error bar for MC accountant is the variance taken over 5 independent runs. Note that the y-axis is in the log scale. (b) Compares the cumulative runtime for online privacy accounting. We did not show AFA [44] as it does not terminate in 24 hours.

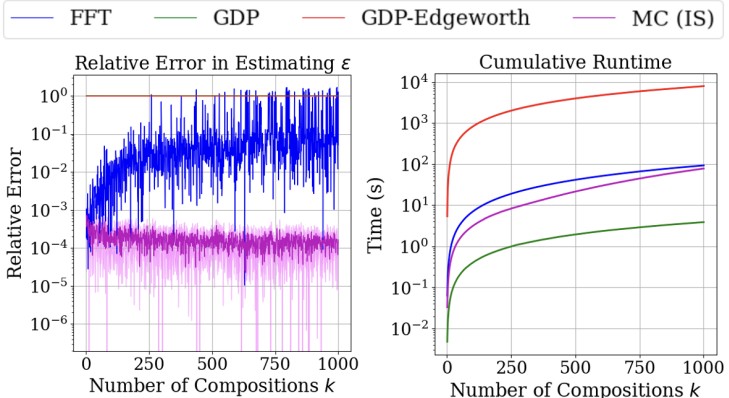

Figure 13: Experiment for Composing Subsampled Gaussian Mechanisms in the Online Setting with hyperparameter $(\sigma, \delta, q) = (0.5, 10^{-13}, 10^{-3})$. (a) Compares the relative error in approximating $k \mapsto \varepsilon_Y(\delta)$. The error bar for MC accountant is the variance taken over 5 independent runs. Note that the y-axis is in the log scale. (b) Compares the cumulative runtime for online privacy accounting. We did not show AFA [44] as it does not terminate in 24 hours.

### H.3.3    Additional Results for Offline Accounting

In this experiment, we set the number of samples for MC accountant as $10^7$, and the parameter for FFT-based method as $\varepsilon_{\mathrm{error}} = 10^{-3}, \delta_{\mathrm{error}} = 10^{-10}$. The parameters are controlled so that the MC accountant is faster than FFT-based method, as shown in Table 1. Figure 14 (a) shows the offline accounting results for $\varepsilon \mapsto \delta_{Y^{(k)}}(\varepsilon)$ when we set $(\sigma, q, k) = (0.5, 10^{-3}, 1000)$. As we can see, the performance of MC accountant is comparable with the state-of-the-art FFT method. In Figure 15 (a), we decreases $q$ to $10^{-5}$. Compared against baselines, MC approximations are significantly more accurate for larger $\varepsilon$, compared with the FFT accountant. Figure 14 (b) shows the offline accounting results for $k \mapsto \varepsilon_Y(\delta)$ when we set $(\sigma, q, \delta) = (0.5, 10^{-3}, 10^{-5})$. Similarly, MC accountant performs comparably as FFT accountant. However, when we decrease $q$ to $10^{-5}$ and $\delta$ to $10^{-14}$ (Figure 15 (b)), MC accountant significantly outperforms FFT accountant. This illustrates that MC accountant performs well in all regimes, and is especially more favorable when the true value of $\delta_Y(\varepsilon)$ is tiny.

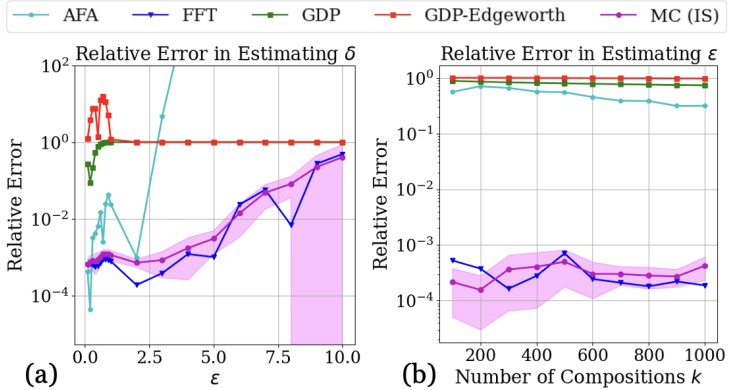

Figure 14: Experiment for Composing Subsampled Gaussian Mechanisms: (a) Compares the relative error in approximating $\varepsilon \mapsto \delta_{Y^{(k)}}(\varepsilon)$ where we set $\sigma = 0.5, k = 1000, q = 10^{-3}$. (b) Compares the relative error in $k \mapsto \varepsilon_Y(\delta)$ where we set $\sigma = 0.5, \delta = 10^{-5}, q = 10^{-3}$. The error bar for MC accountant is the variance taken over 5 independent runs. Note that the y-axis is in the log scale.

Table 1: Runtime for $k = 1000$.

| AFA | GDP | GDP-E | FFT | MC-IS |
|---|---|---|---|---|
| 18.63 | $4.1 \times 10^{-4}$ | 1.50 | 3.01 | 2.31 |

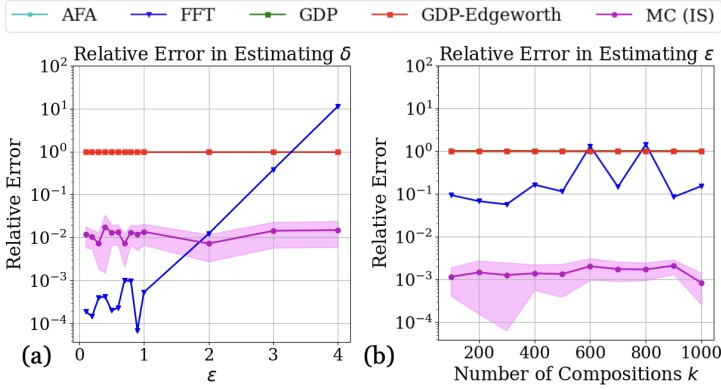

Figure 15: Experiment for Composing Subsampled Gaussian Mechanisms: (a) Compares the relative error in approximating $\varepsilon \mapsto \delta_{Y^{(k)}}(\varepsilon)$ where we set $\sigma = 0.5, k = 100, q = 10^{-5}$. (b) Compares the relative error in $k \mapsto \varepsilon_Y(\delta)$ where we set $\sigma = 0.5, \delta = 10^{-14}, q = 10^{-5}$. The error bar for MC accountant is the variance taken over 5 independent runs. Note that the y-axis is in the log scale. The curves for AFA, GDP and GDP-Edgeworth are overlapped with each other.

