# OpenReview forum: "A Randomized Approach to Tight Privacy Accounting"
_NeurIPS.cc/2023/Conference — NeurIPS 2023 poster_

### Official Review · Reviewer_eMPH · 2023-06-15

**Soundness:** 4 excellent
**Presentation:** 2 fair
**Contribution:** 3 good
**Rating:** 7
**Confidence:** 3

**Summary:**

The authors consider the problem of bounding the privacy loss for a composition of DP mechanisms. This problem is well-studied in the literature and the particular setting here is when the mechanisms are Gaussian or have Gaussian sub-routines but the privacy loss is measured through the original (approximate) DP definition. This is particularly difficult for Gaussians because of the sub-exponential tail and other work has created new definitions better suited to the Gaussian mechanism. While the privacy loss random variable may be challenging to work with under the original DP definition, it can still be efficiently sampled from, so an estimate of the expected privacy loss can be computed efficiently. The authors show that these estimates are quite accurate with high probability and leverage this guarantee to turn the estimate into a formal privacy bound (or failure to run with tiny probability). As a result, for certain settings the authors give improved privacy composition bounds over the previous work.

**Strengths:**

1. Clever new technique to use composition privacy loss estimates instead by rejecting the estimate with low probability.

2. Significant technical work that combines a variety of techniques.

3. Their work extends nicely to the subsampled Gaussian mechanism for DP-SGD.

4. Empirical testing run in a variety of settings for both Gaussian mechanism and DP-SGD.

5. Improves upon previous work for DP composition in a reasonably important setting.

**Weaknesses:**

Remark 6 feels exaggerated. For the settings considered in this paper, the delta parameter is just a result of concentration bounds and sub-exponential tails of Gaussians, not the probability of a catastrophic data breach. Unless the authors know of examples with delta <= 10^{-10}, I've only seen practical applications with delta at minimum 10^{-7} and 10^{-6} is most common. Of course there are privacy advocates that will always push for more privacy, but they are also likely to further push for "pure" differential privacy in which composition is easy and Laplace mechanism must be used instead.

$\textit{The authors pointed to the discussion here I missed, so strike this comment}$
There are privacy definitions better suited to Gaussians that have become very common both in practice and in the literature that allow for easier analysis and make this more difficult accounting less necessary (though still interesting).

$\textit{The authors adequately addressed this in the rebuttal}$
The improvement in epsilon for composition of the Gaussian mechanism over the previous work is quite small (<1%) for their empirical study (figure 5).

**Questions:**

$\textit{All questions were answered by the authors}$

What's the runtime for the 'Exact' method compared to FFT or EVR? It was a bit confusing why that couldn't be used given that it's the ideal privacy parameter to output.

From the experimental result with DP-SGD, it looks like MC beats FFT even with more standard delta settings. Does this hold more generally? Why was this not the result emphasized in the intro instead of the gaussian mechanism composition? Are the curves in figure 4 identical but shifted? So basically using MC accounting you compute a better epsilon at each epoch?

**Limitations:**

The limitations were appropriately discussed

---

> ### Author Rebuttal · Authors · 2023-08-08
>
> We thank the reviewer for the positive comments!
>
> **Q1 [The Importance of Privacy Accounting in Regime of small $\delta$]**
>
> **A:** **Regarding the common rule of setting $\delta = 1/n$:** Setting $\delta$ around $1/n$ implies a maximum $1/n$ likelihood of a privacy guarantee failing for an individual. With a basic union bound, this offers no assurance that the privacy won't fail for everyone in the dataset. Mainstream DP textbooks advocate for a "cryptographically small" $\delta$, specifically, $\delta < n^{-\omega(1)}$ [[1]'s page 25, [2]'s page 9]. The current literature tends to choose a higher value of $\delta$ for a decent utility. Our Remark 6 is intended to remind the community about the importance of using a smaller $\delta$. While it still requires a lot of effort for achieving a good privacy-utility tradeoff even for the current choice of $\delta$, it is important to keep such a goal in mind.
>
> Additionally, recent literature [3] in DP foundation models uses a $\delta$ around $2*10^{-9}$. We anticipate $\delta<10^{-10}$ will become necessary given the trend of using large-scale datasets.
>
> **Our EVR algorithm has improvement when $delta=10^{-5}$ too**. While the improvement for bounding $\delta$ is barely visible when $\delta$ is around $10^{-5}$, the corresponding improvement for $\varepsilon$ can in fact be significant if we align $\delta$ to be the same for the EVR and the original FFT accountant. For example, in Figure 4 we obtain around 0.1 improvements in $\varepsilon$, which lead to around 0.8% accuracy improvement in the privacy-utility tradeoff. Figure 10 in Appendix shows $\varepsilon$-$k$ curves when fixing $\delta = 10^{-5}$. We can see that there’s a non-trivial improvement in $\varepsilon$.
>
> [1] Dwork and Roth. "The algorithmic foundations of differential privacy."
>
> [2] Vadhan, Salil. "The complexity of differential privacy."
>
> [3] Yu et al. “ViP: A Differentially Private Foundation Model for Computer Vision”
>
>
>
> **Q2.** *“There are privacy definitions better suited to Gaussians ...”*
>
> **A:** May we ask the specific privacy definition that *“suited to Gaussians”* the reviewer is referring to? If the reviewer is referring to GDP or RDP (zCDP), we have already shown the disadvantage of the privacy accounting techniques based on these two alternative privacy notions in Figure 1:
>
> **GDP-based accountant:** We can see that GDP accountant completely fails due to the relatively small number of composed mechanism (1200). The original paper [1]’s Figure 3 also shows that GDP's bound is very loose when $m$ is relatively small.
>
> **RDP-based accountant (Moment accountant):** We can see that the Moment Accountant (MA) in Figure 1 is sub-optimal for the entire regime of $\delta$. Moment Accountant is sub-optimal even for the Gaussian mechanism due to the lossy RDP-DP conversion [2].
>
> [1] Analytical composition of differential privacy via the edgeworth accountant. arXiv 2022.
>
> [2] Optimal accounting of differential privacy via characteristic function. AISTATS 2022.
>
>
> **Q3** “The improvement in epsilon for composition of the Gaussian mechanism over the previous work is quite small (<1%) for their empirical study (figure 5).”
>
> **A:** While we agree the absolute improvement is small, we would like to stress that **(1)** Even a slightly more accurate estimation of $\varepsilon$ could result in hundreds of additional training iterations in DP-SGD, which can lead to higher utility; **(2)** Figure 5 aims to illustrate that MC accountant is **both** more accurate and efficient. In other words, MC accountant can achieve a better performance while being around 5 times faster than FFT.
>
> For completeness, we also tune the hyperparameter ($\varepsilon_{error}$) of the FFT accountant to adjust the runtime between FFT and MC accountant to be closer (by setting $\varepsilon_{error}=0.4$ while keeping all other hyperparameters the same). The result is shown in **Q3 in global response & Figure 14 in rebuttal’s PDF**. As we can see, the error of FFT is larger in this case.
>
> **Q4 [How does the `exact’ curve being computed in Figure 1 and 3(a)?]**
>
> **A:** In Figure 1 and 3(a), the “exact” curve can be analytically derived since it’s the composition of pure Gaussian mechanisms (e.g., see [1]’s Equation (10)). This means that the exact $(\varepsilon, \delta)$ curve can be analytically computed, and the composition of pure Gaussian mechanisms mainly served as a toy example for easier performance comparison between different DP accountants. This is a common experiment strategy in the prior literature (e.g., [1]’s Figure 2, [2]’s Figure 2).
>
> [1] Numerical composition of differential privacy. NeurIPS 2021
>
> [2] Connect the Dots: Tighter Discrete Approximations of Privacy Loss Distributions. PETS 2022.
>
> **Q5 [EVR also has improvement on standard $\delta$ setting?]** *“From the experimental ... at each epoch?”*
>
> **A:** In Figure 4, we are comparing between FFT accountant and the FFT accountant augmented with EVR paradigm. That is, we are essentially comparing between FFT accountant’s "upper bound" and "estimate" for epsilon by aligning $\delta$ to be the same, which makes the curves look identical but shifted. In Figure 1, while the improvement for bounding $\delta$ can be barely visible when delta is around $10^{-5}$, the corresponding improvement for $\varepsilon$ can in fact be significant if we align $\delta$ to be the same for the EVR and the original FFT accountant. Hence, in Figure 4 we obtain around 0.1 improvements in $\varepsilon$, which lead to around $0.8\%$ accuracy improvement in the privacy-utility tradeoff curve. Additionally, Figure 10 in Appendix shows $\varepsilon$-$k$ curves when fixing $\delta = 10^{-5}$. We can see that there’s a non-trivial improvement in $\varepsilon$.
>
> We thought Figure 1 is better for illustrating the failure case of “strict upper bound”; for revision, we have further emphasized the improvement of EVR at the regime of $\delta=10^{-5}$ in the Introduction.

---

> > ### Comment · Reviewer_eMPH · 2023-08-10
> >
> > I appreciate the authors thoroughly addressing all comments and questions. Also my apologies for fixating a bit too much on the small delta remark and not taking as much time to better understand other aspects that the authors explained well in the rebuttal.
> >
> > I still feel that in practice setting delta that small will be quite uncommon, and in my opinion that general rule of thumb has become rather antiquated in the same way that experts early on advocated for $\epsilon << 1$ which is generally far too impractical for industry use-cases. But I appreciate the authors clarifying the other points and emphasizing the improvement in larger delta regimes for future versions of their work.

---

> > > ### Author Response · Authors · 2023-08-11
> > > **Thanks for the prompt response!**
> > >
> > > We sincerely thank the reviewer for the prompt response and for raising the score! We will incorporate your comments about the consideration of $\delta$ for practical industry use-cases into the revision!

---

### Official Review · Reviewer_hkgF · 2023-07-05

**Soundness:** 3 good
**Presentation:** 3 good
**Contribution:** 3 good
**Rating:** 7
**Confidence:** 4

**Summary:**

Authors introduce a new privacy accounting method to characterize the privacy loss random variable. The work reduce the classical privacy accounting problem into mean estimation problem following the previous work and give a Monte Carlo solution.  The work provides detailed analysis of the proposed method and its utility and also give some common distribution to show the effectiveness and performance theoretically. The numerical studies also show the correctness and practicability of the method in the real privacy accounting tasks.

**Strengths:**

 Pros:
1. The analysis for the proposed method is detailed and the writing is friendly to follow with a detailed preliminary.
2. The proposed tool works better than the compared existing accounting tools like CLT and FFT-based methods.
3. The fast speed and the online implementation show some potential for the method used in privacy accounting applications.



**Weaknesses:**

Actually, I have reviewed the paper in ICML. I think the paper has fixed almost the problem in the former cycle. The only problem I still keep is about whether the method will suffer from the dimension curse when deriving the prv samples for a general distribution.

**Questions:**

/

**Limitations:**

Yes

---

> ### Author Rebuttal · Authors · 2023-08-09
>
> We thank the reviewer (again) for the very positive feedback!
>
> **Q** *“The only problem I still keep is about whether the method will suffer from the dimension curse when deriving the prv samples for a general distribution.”*
>
> **A:** This is a great comment that actually points to one of the important benefits of our approach. The cost of our approach would not increase (in terms of number of samples) even if the dominating pair is supported in a high dimensional space. For Monte Carlo estimate $\delta = 1/m \sum_{i=1}^m (1-e^{\varepsilon - y_i})_+$, $y_i$ is the sampled log density ratio, and hence it is a scalar value. We can now see from Hoeffding’s inequality that the expected error rate of estimation is $O(1/\sqrt{m})$ which is independent of the dimension of the support set of dominating distribution pairs. This means that the number of samples we need to ensure a certain confidence interval is independent of the dimension. However, we should also note that although the number of samples does not change, the sampling process itself might be more costly for higher dimensional spaces. But one would expect that to grow at most polynomially. We will clarify this in the paper.

---

### Official Review · Reviewer_UAod · 2023-07-05

**Soundness:** 3 good
**Presentation:** 3 good
**Contribution:** 3 good
**Rating:** 6
**Confidence:** 3

**Summary:**

The paper proposes a privacy accounting method called estimate-verify-release (EVR), whose basic principle is to convert an estimate of a privacy parameter into a formal privacy guarantee. The mechanism works by verifying whether the estimated privacy guarantee holds, and then releasing the query output depending on the verification result. The paper develops a Monte-Carlo-based verifier for this paradigm. The overall accountant is broadly applicable and is shown to give a tighter privacy-utility tradeoff than existing baselines.

**Strengths:**

The mechanism is applicable broadly, in particular to important DP algorithms such as DP-SGD (that is, the subsampled Gaussian mechanism). It exploits the fact that existing work provides good privacy loss estimates and converts these estimates into a formal paradigm for ensuring differential privacy. The method is shown to beat a strong baseline, namely the FFT-based accountant from [19].

**Weaknesses:**

The paper places a lot of emphasis on the fact that we can't naively use an estimated privacy parameter as the truth, because DP is a strict guarantee, and this makes perfect sense. But then in the analysis and implementation of the accountant there are some steps of the new accountant that are not made completely rigorous, such as the number of Monte Carlo samples. Or, in Theorem 13 there is a nu parameter that is not known. So my suggestion is to write a fully formal version of the accountant for DP-SGD in the main body to show that the paradigm can indeed be applied fully rigorously.

In addition, it would be good to see at least one more experiment showing the same comparison as Figure 5. In Figure 5 the gains of MC over FFT are not clear for a small number of compositions. So it would be good to see if the comparison of Figure 5 is robust and generalizes to different problem settings.

A few minor comments:
- In the Conclusion, you say "allowing safe privacy parameter estimates *without* provable assurance"?
- The Figures don't seem to be in vector format and are blurry if zoomed in.
- Typo in Line 132, "privacy loss random variable *is* ..." (right now there's no verb in the main clause)
- In Line 180, say where rho takes values.

**Questions:**

Nothing at the moment.

---

> ### Author Rebuttal · Authors · 2023-08-09
>
> We thank the reviewer for the very positive feedback!
>
> **Q1 [Fully formal version of DP-SGD with EVR paradigm]** *“The paper places a lot of emphasis on the fact that we can't naively use an estimated privacy parameter as the truth, because DP is a strict guarantee, and this makes perfect sense. But then in the analysis and implementation of the accountant there are some steps of the new accountant that are not made completely rigorous, such as the number of Monte Carlo samples. Or, in Theorem 13 there is a nu parameter that is not known. So my suggestion is to write a fully formal version of the accountant for DP-SGD in the main body to show that the paradigm can indeed be applied fully rigorously.”*
>
> **A:** We thank the reviewer for the very concrete and useful suggestion. In the updated paper, we have included a pseudo-code outlining the full steps of DP-SGD with the EVR paradigm, as well as a theorem stating that the algorithm satisfies DP guarantee (**see Q1 in global response and Algorithm 4 in the rebuttal’s PDF**).
>
> For ***“in Theorem 13 there is a nu parameter that is not known”:*** $\nu$ is the upper bound for the second moment of the MC estimator, which we discuss how to bound it (which can be explicitly computed) for $\delta_{SMC}$ and $\delta_{IS}$ in the last paragraph of Section 4.3 and Appendix E.
>
> **Q2 [More experiment showing the same comparison as Figure 5]** *“In addition, it would be good to see at least one more experiment showing the same comparison as Figure 5. In Figure 5 the gains of MC over FFT are not clear for a small number of compositions. So it would be good to see if the comparison of Figure 5 is robust and generalizes to different problem settings.”*
>
> **A:** Thanks for the suggestion! We have conducted additional experiment in Appendix G.3.2, and the results are shown in **Figure 12 and 13 in the rebuttal’s PDF (see Q2 in Global response)**. The experiment settings follow exactly the same as Figure 5. Figure 12 and 13 show the online accounting results for $(\sigma, \delta, q) = (0.5, 10^{-5}, 10^{-3})$ and $(\sigma, \delta, q) = (0.5, 10^{-13}, 10^{-3})$, respectively. For the setting of $(\sigma, \delta, q) = (0.5, 10^{-5}, 10^{-3})$, we can see that the MC accountant achieves a comparable performance with a shorter runtime. For the setting of $(\sigma, \delta, q) = (0.5, 10^{-13}, 10^{-3})$, we can see that the MC accountant achieves significantly better performance compared to the state-of-the-art FFT accountant (and again, with a shorter runtime). This further showcases the MC accountant's efficiency and accuracy in online setting.
>
> **Q3 [Typos & Grammar & Blurred Images]**
>
> **A:** Thanks a lot for the catch! We have fixed the typos and grammars in the paper. For the comment about blurred image, we checked all of the images and they all look clear even after zoomed in on our side. Could you kindly point out the specific figure you are talking about? We are more than happy to change it!

---

> > ### Comment · Reviewer_UAod · 2023-08-12
> > **Thank you**
> >
> > Thank you for the response; it was helpful and I am happy to see that all my concerns are addressed. Regarding the figures, if you zoom in all the way the figures look pixelated because they are raster (such as png, jpg) and not vector (such as pdf). In my opinion it does not look professional to have figures that are raster in a paper.

---

> > > ### Author Response · Authors · 2023-08-12
> > > **Thanks for the response!**
> > >
> > > We sincerely thank the reviewer for the positive feedback! We will definitely change the figures' format according to your suggestion for the revision!

---

### Official Review · Reviewer_D6i7 · 2023-07-07

**Soundness:** 4 excellent
**Presentation:** 4 excellent
**Contribution:** 3 good
**Rating:** 7
**Confidence:** 4

**Summary:**

The authors propose EVR framework for privacy accounting. The core idea is to estimate the privacy budget, verify whether the budget is approximately met, and then decide whether to release the result or halt. The workhorse is a Monte-Carlo verifier (also used as an accountant through binary search). The empirical evaluation shows significant improvement over existing techniques, especially in the large epsilon/small delta regime.

**Strengths:**

1. The authors propose a novel framework for privacy accounting: EVR. By trading off a small probability that the program halts with privacy budget, the authors manage to get a tighter privacy profile curve at larger epsilon/smaller delta regime.
2. The authors verify the proposed framework empirically and showcase the advantage of EVR.

**Weaknesses:**

1. One downside of the proposed approach is that there is a probability that the mechanism halts with a privacy budget cost. To get a tighter epsilon, you need to take the risk that you get nothing. Although this seems to be reasonable trade-off, it can be a problem in practical use case. More discussion should be put on this.

**Questions:**

Any idea on how to deal with halting in practical usage?

**Limitations:**

Yes

---

> ### Author Rebuttal · Authors · 2023-08-07
>
> We thank the reviewer for the very positive feedback!
>
> **Q [How to deal with halting in practical usage?]**
>
> **A:** In Section 4.4, we developed techniques for ensuring the false negative rate (i.e., the rejection probability) is around O(delta) when the proposed privacy parameter delta^est is close to the true delta. In Section 6.1’s experiment, we use the state-of-the-art FFT accountant to produce delta^est, which is very accurate as we can see from Figure 1. Hence, the rejection probability in the experiment is around O(delta), which means the probability of rejection is close to the probability of catastrophic failure for privacy.
>
> In addition, if one is still concerned that the rejection probability is too large, we can further reduce the probability as follows: we run two instances of EVR paradigm simultaneously; if both of the instances are passed, we randomly pick one and release the output. If either one of them is passed, we release the passed instance. It only fails when both of the instances fail. By running two instances of the EVR paradigm in parallel, the false positive rate (i.e., the final $\delta$) will be only doubled, but the probability of rejection will be squared.
>
> We can also introduce a variant of our EVR paradigm that better deals with the failure case: whenever we face the "rejection", we run a different mechanism $M’$ that is guaranteed to be $(\epsilon, \delta^{(est)})$-DP (e.g., by adjusting the subsampling rate and/or noise multiplier in DPSGD). Moreover, we use FFT accountant to obtain a strict privacy guarantee upper bound $(\epsilon, \delta^*)$ for the original mechanism $M$, where $\delta^{(est)} < \delta^*$. We use $p_{fp}$ and $p_{fn}$ to denote the false positive and false negative rate of the underlying DP verifier.
> - If the original mechanism $M$ is indeed $(\epsilon, \delta^{(est)})$-DP, then for any subset $S$ we have
>
> $\Pr[EVR(D) \in S] = p_{fn} * \Pr[M(D) \in S] + (1-p_{fn}) * \Pr[M’(D) \in S] $
>
> $\ \ \ \ \ \ \ \ \ \ \ \ \ \ \ \ \ \ \ \ \ \ \ \ \ \ \ \ \ \     \le p_{fn} * (e^\epsilon \Pr[M(D’) \in S] + \delta^{(est)} ) + (1-p_{fn}) * (e^\epsilon \Pr[M’(D’) \in S] + \delta^{(est)}) $
>
> $\ \ \ \ \ \ \ \ \ \ \ \ \ \ \ \ \ \ \ \ \ \ \ \ \ \ \ \ \ \  \le e^\epsilon \Pr[EVR(D) \in S] + \delta^{(est)}$
>
> - If the original mechanism $M$ is not $(\epsilon, \delta^{(est)})$-DP, then we have
>
> $\Pr[EVR(D) \in S] = p_{fp} * \Pr[M(D) \in S] + (1-p_{fp}) * \Pr[M’(D) \in S]$
>
> $\ \ \ \ \ \ \ \ \ \ \ \ \ \ \ \ \ \ \ \ \ \ \ \ \ \ \ \ \ \  \le p_{fp} (e^\epsilon \Pr[M(D’) \in S] + \delta^*) + (1-p_{fp}) * (e^\epsilon \Pr[M’(D’) \in S] + \delta^{(est)})$
>
> $\ \ \ \ \ \ \ \ \ \ \ \ \ \ \ \ \ \ \ \ \ \ \ \ \ \ \ \ \ \  \le e^\epsilon \Pr[EVR(D) \in S] + \delta^{(est)} + p_{fp} * (\delta^* - \delta^{(est)})$
>
> Hence, this augmented EVR algorithm will be $(\epsilon, \delta^{(est)} + p_{fp} * (\delta^*-\delta^{(est)}))$-DP, and if $p_{fp}$ is around $\delta^{(est)}$, then this extra factor $p_{fp} * (\delta^*-\delta^{(est)})$ will be very small. We can also adjust the privacy guarantee for $M’$ such that the privacy guarantees for the two cases are the same, which can further optimize the final privacy cost.
>
> We have added the above discussion to the paper.

---

### Author Rebuttal · Authors · 2023-08-09

We thank all of the reviewers for their detailed and valuable comments. We are pleased that all the reviewers expressed a positive view of our work! We considered the reviews carefully and modified our paper accordingly. We have answered other questions in the individual responses. Here’s a summary of the contents in the submitted PDF:

**Q1 (for Reviewer UAod) [Fully formal version of DP-SGD with EVR paradigm]**

**A:** We thank Reviewer UAod for the very concrete and useful suggestion. Here, we additionally outline the full steps of privacy accounting for DP-SGD with our EVR paradigm through pseudo-code in **Algorithm 4 in Rebuttal’s pdf**. Recall from Lemma 16 that for subsampled Gaussian mechanism with sensitivity $C$, noise variance $C^2 \sigma^2$, and subsampling rate $q$, one dominating pair $(P, Q)$ is $Q := N(0, \sigma^2)$ and $P := (1-q) N(0, \sigma^2) + q N(1, \sigma^2)$. Hence, for DP-SGD with $k$ iterations, the dominating pair is the product distribution **P** $ := P_1 \times \ldots \times P_k$ and **Q** $:= Q_1 \times \ldots \times Q_k$ where each $P_i$ and $Q_i$ follow the same distribution as $P$ and $Q$ (It can also be extended to the heterogeneous case easily).

We have also included the following corollary in the paper so that the paradigm can indeed be applied fully rigorously.

**Corollary:** The Step 2-3 in Algorithm 4 is $(\varepsilon, \delta^{est}/\tau)$-DP.

The corollary directly follows from Theorem 9 in the paper.

**Q2 (for Reviewer UAod) [More experiments for evaluating MC accountant in online setting]**

**A:** Thanks for the suggestion! We have conducted additional experiment in Appendix G.3.2, and the results are shown in **Figure 12 and 13 in the rebuttal’s PDF**. The experiment settings follow exactly the same as Figure 5. Figure 12 and 13 show the online accounting results for $(\sigma, \delta, q) = (0.5, 10^{-5}, 10^{-3})$ and $(\sigma, \delta, q) = (0.5, 10^{-13}, 10^{-3})$, respectively. For the setting of $(\sigma, \delta, q) = (0.5, 10^{-5}, 10^{-3})$, we can see that the MC accountant achieves a comparable performance with a shorter runtime. For the setting of $(\sigma, \delta, q) = (0.5, 10^{-13}, 10^{-3})$, we can see that the MC accountant achieves significantly better performance compared to the state-of-the-art FFT accountant (and again, with a shorter runtime). This further showcases the MC accountant's efficiency and accuracy in online setting.

**Q3 (for Reviewer eMPH) [Additional experiments which increase the runtime for FFT accountant]**

**A:** Figure 5 aims to illustrate that MC accountant is **both** more accurate and efficient. In other words, MC accountant can achieve a better performance while being around 5 times more efficient than FFT. To better illustrate the advantages of MC accounting over the FFT accountant, we tune the hyperparameter ($\varepsilon_{error}$) of the FFT accountant so that the runtime of FFT and MC accountant are closer to each other. Specifically, we set $\varepsilon_{error}=0.4$ while keeping all other hyperparameters to be the same. The result is shown in **Figure 14 in the rebuttal’s PDF**. As we can see, the improvement of the MC accountant over the FFT accountant is larger in this case.

---

### Decision · Program_Chairs · 2023-09-21

**Decision:**

Accept (poster)

**Comment:**

There was a consensus among reviewers to accept this paper. The proposed approach is a nice, principled and useful addition to the privacy accounting literature, which gives tighter bounds than existing methods while being reasonably efficient. Therefore, I recommend acceptance.